# Stretching muscle cells induces transcriptional and splicing transitions and changes in SR proteins

Emma R. Hinkle [1,2], R. Eric Blue [1], Yi-Hsuan Tsai [3], Matthew Combs[4], Jacquelyn Davi[1], Alisha R. Coffey[3], Aladin M. Boriek[5], Joan M. Taylor [4,6], Joel S. Parker[2,3] & Jimena Giudice [1,2,6 ✉]

Alternative splicing is an RNA processing mechanism involved in skeletal muscle development and pathology. Muscular diseases exhibit splicing alterations and changes in mechanobiology leading us to investigate the interconnection between mechanical forces and RNA processing. We performed deep RNA-sequencing after stretching muscle cells. First, we uncovered transcriptional changes in genes encoding proteins involved in muscle function and transcription. Second, we observed that numerous mechanosensitive genes were part of the MAPK pathway which was activated in response to stretching. Third, we revealed that stretching skeletal muscle cells increased the proportion of alternatively spliced cassette exons and their inclusion. Fourth, we demonstrated that the serine and arginine-rich proteins exhibited stronger transcriptional changes than other RNA-binding proteins and that SRSF4 phosphorylation is mechanosensitive. Identifying SRSF4 as a mechanosensitive RNA-binding protein that might contribute to crosstalk between mechanotransduction, transcription, and splicing could potentially reveal novel insights into muscular diseases, particularly those with unknown etiologies.

[1] Department of Cell Biology and Physiology, The University of North Carolina at Chapel Hill, Chapel Hill 27599 NC, USA. [2] Curriculum in Genetics and Molecular Biology (GMB), The University of North Carolina at Chapel Hill, Chapel Hill 27599 NC, USA. [3] Lineberger Comprehensive Cancer Center, The University of North Carolina at Chapel Hill, Chapel Hill 27599 NC, USA. [4] Department of Pathology and Laboratory Medicine, The University of North Carolina at Chapel Hill, Chapel Hill 27599 NC, USA. [5] Department of Medicine, Baylor College of Medicine, Houston, TX, USA. [6] McAllister Heart Institute, The University of North Carolina at Chapel Hill, Chapel Hill 27599 NC, USA. ✉email: jimena_giudice@med.unc.edu

Alternative splicing is an RNA processing mechanism that expands protein diversity and is extremely prevalent in higher eukaryotes with over 95% of human genes undergoing alternative splicing[1]. In skeletal muscle, extensive alternative splicing transitions occur postnatally[2] contributing to the maturation of the contractile apparatus[3–6]. Skeletal muscle is a complex tissue dependent on both molecular mechanisms and mechanical properties for its development. The muscle contractile apparatus consists of sarcomeres, transverse tubules, and costameres, which together transduce force both across muscle cells and from the nucleus to the plasma membrane in a process known as mechanotransduction[7]. Interestingly, in various muscular diseases, adult alternative splicing patterns revert to fetal stages contributing to muscle loss, atrophy, and functional failure of the muscle[3,4,8,9]. Thus, molecular transitions in skeletal muscle are important for proper development of the tissue.

Besides undergoing molecular transitions during development, skeletal muscle must respond to mechanical forces and generate coordinated bursts of force through contraction. Muscle cells are particularly sensitive to mechanotransduction[10]; the stiffness of their surroundings controls cell elongation and differentiation, which in turn are critical for efficient muscle contraction[11]. In comparison with healthy individuals, muscle cells from those suffering from muscular dystrophy are stiffer and this is thought to prevent proper contractility – ultimately leading to muscle wasting[11–13].

It has been extensively demonstrated that global missplicing or misregulation of individual alternative splicing events results in the loss of proper muscle function[2–4,6,14,15]. Mice with Duchenne muscular dystrophy (which dysregulates splicing) present alterations in mechanosensitive signaling pathways indicating that physical changes in muscle are linked to molecular responses[16]. Expanding upon this idea, aging mice with muscular dystrophy exhibit reduced fatigue in response to an intervention that includes both splice-switching therapy and exercise[17]. These studies establish a link between muscular diseases, splicing misregulation, and mechanical alterations of the muscle. However, it is unknown how mechanical forces and alternative splicing cooperate to maintain muscle homeostasis and to what extent they contribute to the onset of muscular diseases.

In this study, we investigate how stretching skeletal muscle cells impacts global alternative splicing and gene expression programs using deep RNA-sequencing (RNA-seq). We discovered that mechanical stretching induces extensive transcriptional changes in genes encoding proteins involved in transcription, muscle cell differentiation, and the mitogen-activated protein kinase (MAPK) pathway. Further, stretching promotes alterations in the splicing of cassette exons and favors exon inclusion over skipping. This evidence suggests that RNA-binding proteins (RBPs) might play a role in inducing those splicing changes in response to cellular stretching.

## Results

### Stretching induces gene expression changes in undifferentiated and differentiated muscle cells

Various studies have examined the effect of stretching on signaling pathways, but to our knowledge, no study has utilized an unbiased and global approach to define how stretching impacts gene expression and alternative splicing programs in skeletal muscle cells. To address this gap of knowledge, we utilized the Flexcell system to apply mechanical forces to muscle cells for different periods of time (1 h, 3 h, and 6 h) and then performed deep RNA-seq studies.

We opted to perform our study in two different stages of muscle cell differentiation (myogenesis) to determine how stretching impacted the transcriptional and posttranscriptional

programs that were unique to these stages. During development, muscle matures the intracellular architecture required for proper adult contraction. Myoblasts are undifferentiated muscle cells that mimic early muscle development or muscle that has atrophied due to disease; while differentiated muscle cells more closely mimic normal, and more adult muscle. Notably, differentiation of C2C12 myoblasts into multinucleated myotubes in culture reproduces the transcriptional and posttranscriptional transitions observed during skeletal muscle development in vivo[2,18,19] which establishes these cells as a good cell culture model for understanding mechanotransduction[18–20]. We stretched undifferentiated cells (myoblasts) and cells after four days of differentiation (differentiated cells).

Most cell stretching systems can only apply mechanical tension in one direction which is not an accurate model for skeletal muscle contraction. During contraction, skeletal muscle generates force across myofibers by transferring force both laterally and longitudinally[21]. The Flexcell system that we have utilized here was unique because it stretched cells in a cyclic, equibiaxial manner which is a widely accepted model of in vivo skeletal muscle force transmission[22–24]. The amount of time the cells were stretched was chosen based on previous studies[25] and to investigate an acute response (1 h), an intermediate response (3 h), and a sustained response (6 h) to stretch. We stretched cells with the maximum amount possible with our system (16%) which corresponds to a pressure of −69.5 (kPa) and a strain of 0.16 exerted on the flexible membrane that the cells were plated on.

After sequencing, we obtained a range of 63–103 million read pairs per sample with >90% of them mapping to the mm10 mouse genome (Supplementary Data 1). The high mapping rate indicates appropriate quality of our sequencing data. We first confirmed that several myogenesis markers were not expressed in the myoblast samples and were highly induced in the differentiated cells (Supplementary Fig. 1). We next performed principal component analysis (PCA) to cluster samples based on gene expression for both myoblasts (Fig. 1a) and differentiated cells (Fig. 1b). The non-stretched myoblasts segregated together with two distinct subclusters for the 1 h stretched samples and the 6 h stretched samples (Fig. 1a, *dark green* and *dark blue* respectively). The 3 h stretched myoblasts segregated closer to the non-stretched samples (Fig. 1a, *dark purple*). In the differentiated cells, all the non-stretched samples segregated together and we observed three distinct groups for the different stretching time points demonstrating that varying times of stretching resulted in distinct gene expression changes (Fig. 1b, *dark green, dark purple*, and *dark blue*). One of the 1 h stretched samples did not segregate closely with the other replicates but remained more similar to them than to samples at other time points. This sample passed other quality controls and was therefore included with all samples in the analysis described in the results. *Post hoc* tests excluding this sample were performed and did not change the conclusions of the study.

In myoblasts, there were 151 and 134 genes downregulated after 1 h and 3 h of stretching, respectively, when compared to non-stretched cells while only a few genes (37 and 46, respectively) were upregulated at these time points (Fig. 1c, *left*). After 6 h of stretching, we found 158 upregulated genes compared to non-stretched samples, while 102 genes were downregulated (Fig. 1c, *right*). In differentiated cells, there were 54 and 98 genes downregulated in response to 1 h and 3 h of stretching, respectively, when compared to non-stretched samples. For the stretched, differentiated cells, several genes (44 and 107), were upregulated after 1 h and 3 h of stretching, respectively (Fig. 1d, *left*). After 6 h of stretching we found 188 downregulated genes compared to the non-stretched samples, while 67 genes were

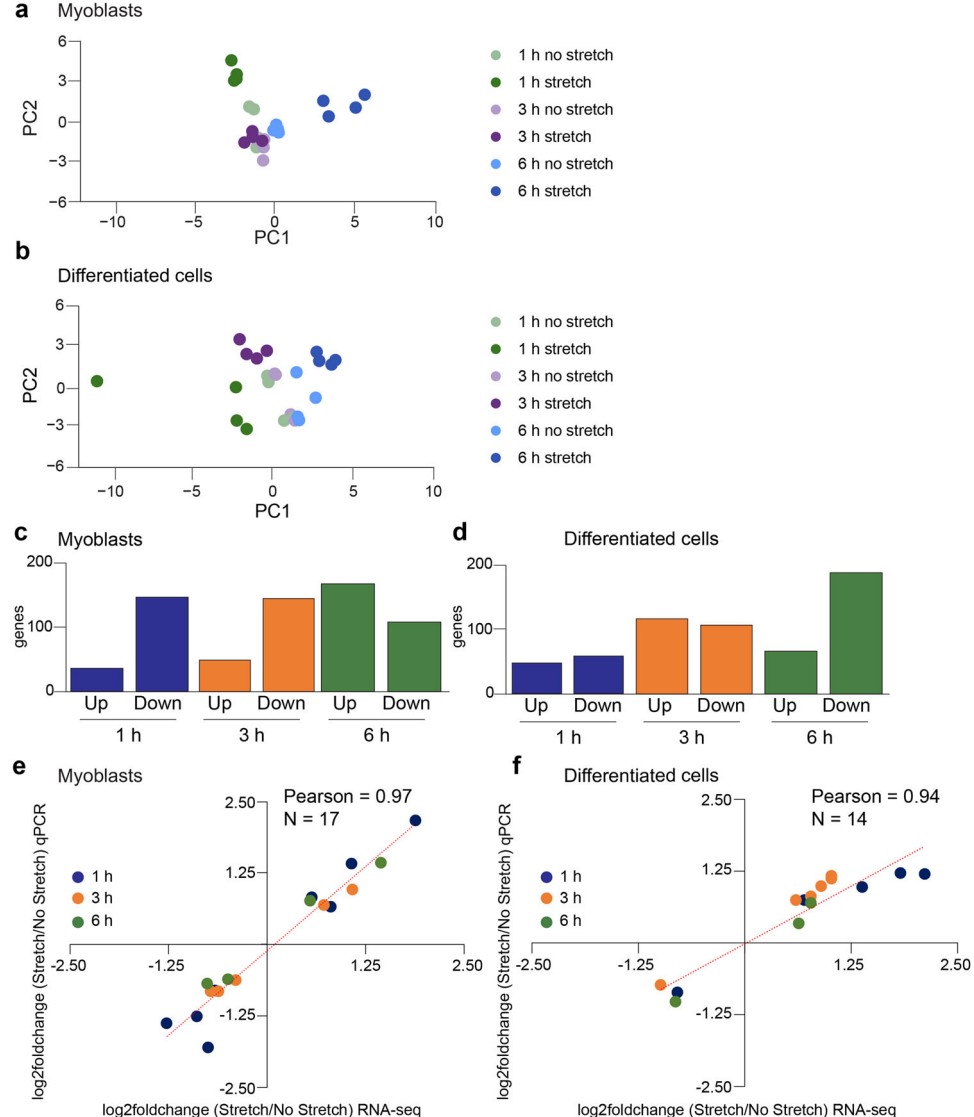

**Fig. 1 Extensive gene expression changes are induced upon stretching of muscle cells. a**, **b** Principal component analysis (PCA) plots of myoblasts and differentiated cells. **c**, **d**. Global gene expression changes after 1 h, 3 h, or 6 h stretching of myoblasts and differentiated cells. Gene expression changes were calculated by comparing stretched and non-stretched samples. Genes were considered differentially expressed when the adjusted $p$-value<0.05 and the fold change >1.50 (upregulated, up) or fold change < −1.50 (downregulated, down). **e**, **f**. Correlation plot between the gene expression changes (expressed as log$_2$foldchange) as measured by qPCR assays and those observed in the RNA-seq studies in myoblasts and differentiated cells. The qPCR graphs for the individual genes included in the correlation plots are shown in Supplementary Fig. 2 (myoblasts) and Supplementary Fig. 3 (differentiated cells).

upregulated (Fig. 1d, *right*). Overall, in both myoblasts and differentiated cells stretching for 1 h and 3 h triggered more similar changes in gene expression than stretching for 6 h.

We validated gene expression changes in response to stretching using quantitative real time PCR (qPCR) in myoblasts (Supplementary Fig. 2) and differentiated cells (Supplementary Fig. 3). We selected genes for validation with a fold change >1.5 for upregulated genes and a fold change <−1.5 for downregulated genes and an adjusted $p$-value < 0.05. Two of the genes assayed in both myoblasts and differentiated cells were the connective tissue growth factor (*Ctgf*) and the cysteine-rich angiogenic inducer 61 (*Cyr61*) which are both well-established mechanosensitive genes[26–28]. *Ctgf* and *Cyr61* mRNAs were upregulated after 1 h of stretching in myoblasts and after 1 h and 3 h of stretching in differentiated cells indicating successful cellular stretching (Supplementary Figs. 2, 3). After 6 h of stretching, *Ctgf* and *Cyr61* mRNAs were not upregulated in either myoblasts or

differentiated cells (Supplementary Figs. 2, 3) suggesting that the cells become accustomed to the long mechanical stimulus which has been previously reported[28].

A correlation plot was generated to compare the mRNA expression changes detected in the RNA-seq studies with those measured by qPCR assays. We observed a strong correlation between the two approaches both in myoblasts (Pearson = 0.97, Fig. 1e) and in differentiated cells (Pearson = 0.94, Fig. 1f). This robust validation establishes our RNA-seq data as high-quality.

**Genes encoding proteins involved in muscle function and transcription are upregulated in response to cell stretching.** We next asked whether the genes that were up- or downregulated at different times of stretching were functionally related. To answer this question, we performed gene ontology (GO) analysis using the Database for Annotation, Visualization and Integrated

Discovery (DAVID). For myoblasts, the genes that responded to 1 h and 3 h of stretching encoded proteins involved in transcription regulation and muscle cell differentiation while categories related to protein folding were evident for genes that responded to a longer period of stretching (6 h) (Supplementary Fig. 4a). In differentiated cells, genes responding to stretching encoded proteins involved in muscle function, transcription, and the transforming growth factor β (TGF-β) signaling pathway (1 h) and steroid homeostasis (3 h and 6 h) (Supplementary Fig. 4b). In both myoblasts and differentiated cells, genes that responded to shorter stretching times tended to encode proteins functioning in transcriptional regulation and muscle differentiation. Overall, our data demonstrates in an unbiased manner that mechanical stretching of skeletal muscle cells induces distinct, global mRNA expression changes in genes encoding proteins involved in cell differentiation and transcription regulation.

**The MAPK pathway is activated after cell stretching**. Since mechanical forces impact the plasma membrane of the cell, we were interested in identifying the specific intracellular signaling pathways that were activated in response to stretching. To investigate this, we performed DAVID pathway analysis on the genes that responded to 1 h or 3 h of stretching in differentiated cells. After 1 h of stretching, the responsive genes were enriched in pathways including the MAPK cascade, epidermal growth factor receptor family (ErBβ), the Janus kinase-signal transducer and activator of transcription (JAK/STAT) pathway, and tumor necrosis factor (TNF) signaling (Fig. 2a). Few pathways were enriched in the samples that were stretched for 3 h. Interestingly, numerous upregulated genes after 1 h of stretching encoded proteins that are part of the extracellular signal-related kinase (ERK1/2) cascade, which is at the intersection of the MAPK, ErBβ, and JAK/STAT axes. ERK1/2 phosphorylation was drastically decreased during C2C12 cell differentiation while total ERK1/2 protein levels did not change (Supplementary Fig. 5). Interestingly, ERK1/2 were significantly phosphorylated in response to 1 h and 3 h of cellular stretching in differentiated cells (Fig. 2b). It was previously demonstrated that ERK1/2 are phosphorylated in response to a stretch stimulus in mouse and rat skeletal muscle as well as C2C12 cells, so our data corroborates previous work[29–31].

To further define the effect of stretching on MAPK signaling, we used our RNA-seq data to examine the activation of various MAPK targets. Numerous genes in this pathway were upregulated after 1 h of cell stretching and slightly upregulated after 3 h of stretching (Fig. 2c). Moreover, when we validated some of these targets by qPCR assays, we confirmed the significant upregulation (Supplementary Fig. 3, *Myc* and *Ctgf*). Collectively, this evidence led us to conclude that short stretching (1 h) activates the MAPK axis to induce the transcription of genes encoding proteins that function in signal transduction and transcriptional activation of other genes (Fig. 2d).

**ERK phosphorylation inhibition impacts the response of mechanosensitive genes**. Since ERK1/2 phosphorylation was mechanosensitive we aimed to determine whether this pathway played a role in activating transcription of genes upon stretching. To do this, we used a well-established ERK1/2 phosphorylation inhibitor (U0126) and stretched differentiated cells for 1 h (with or without the inhibitor). We chose this stretching time since the strongest ERK1/2 phosphorylation occurred after 1 h of stretching (Fig. 2b). First, we confirmed the complete inhibition of ERK1/2 phosphorylation in both the non-stretched and stretched cells after treatment with U0126 (Fig. 3a). We next assayed the mRNA expression levels of several mechanosensitive genes that

are part of the MAPK pathway. For several of these genes, U0126 abolished the response to stretch (Fig. 3b, *Nr4a1*, *Atf3*, and *Ccl2*), indicating that ERK1/2 plays a role in their activation upon stretching. For some other genes, the response to stretching was partially abolished in the presence of U0126 (Fig. 3b, *Ctgf*, *Cyr61*, *Fos*, *Myc*, and *Ereg*) which suggests that ERK1/2 may not be the only activator of these targets upon stretching, but may help sustain their activation. In summary, inhibition of ERK1/2 phosphorylation influenced some mechanosensitive genes indicating a role of this pathway in the stretch response of differentiated cells.

**Splicing changes increase as cells are stretched over time**. RNA processing can be regulated by multiple factors including transcriptional speed, temperature, cellular stress, and other environmental factors[32–34]. We thus hypothesized that mechanical stretching induces alternative splicing changes in skeletal muscle cells.

To examine the distribution of the alternatively spliced events that responded to stretching, we utilized the Mixture of Isoforms (MISO) software[35]. MISO allows accurate determination of alterative splicing patterns at the exon level. We defined the change in percent spliced in (ΔPSI) as the difference between the PSI in stretching conditions and the PSI in non-stretched samples ($PSI_{stretch} - PSI_{no\ stretch}$)[36]. Events were considered differentially spliced if the Wilcox $p$-value $\leq 0.05$ and $|\Delta PSI| > 10$. In myoblasts, we observed an increase in the total number of splicing events as the cells were stretched for longer periods of time (Fig. 4a, *left*). Whereas in differentiated cells, the total number of splicing events increased from 1 h to 3 h and slightly decreased from 3 h to 6 h (Fig. 4a, *right*).

PCA analysis revealed correlations between the alternative splicing events across increasing times of stretching. For myoblasts, we observed distinct segregation for the 1 h, 3 h, and 6 h stretched samples (Fig. 4b, *dark green, dark purple*, and *dark blue*, respectively) and segregation of the non-stretched samples. Similarly, the non-stretched differentiated cells segregated together and the 1 h, 3 h, and 6 h stretched samples exhibited definite groups (Fig. 4c, *dark green, dark purple*, and *dark blue* respectively). These data led us to conclude that stretching induces distinct changes in alternative splicing at each time point.

To validate the splicing transitions detected by RNA-seq in response to stretching, we performed reverse transcription PCR (RT-PCR) analysis of 12 cassette exons in myoblasts (Supplementary Fig. 6) and 11 cassette exons in differentiated cells (Supplementary Fig. 7). Some of these events were assayed across the time points because RNA-seq data revealed that they responded to stretch at different times. We observed a strong correlation (Pearson = 0.82 in myoblasts and Pearson=0.84 in differentiated cells) between the ΔPSI values from RNA-seq and those from RT-PCR experiments (Fig. 4d, e).

Overall, we conclude that our RNA-seq study produced highly reproducible and robust data and that numerous splicing events are mechanosensitive in muscle cells.

**Alternative splicing changes in response to stretching occur mostly in cassette exons**. Different types of splicing events exist including cassette exons, mutually exclusive exons, retained introns, alternative 3' splice sites, and alternative 5' splice sites (Fig. 5a). We thus asked whether there was a type of splicing event that was more prevalent and to what extent the distribution of splicing event types changed over stretching time. First, we observed that >44% of splicing events at all time points were cassette exons (Fig. 5b). Second, we found that as cells were stretched, the proportion of cassette exons being alternatively

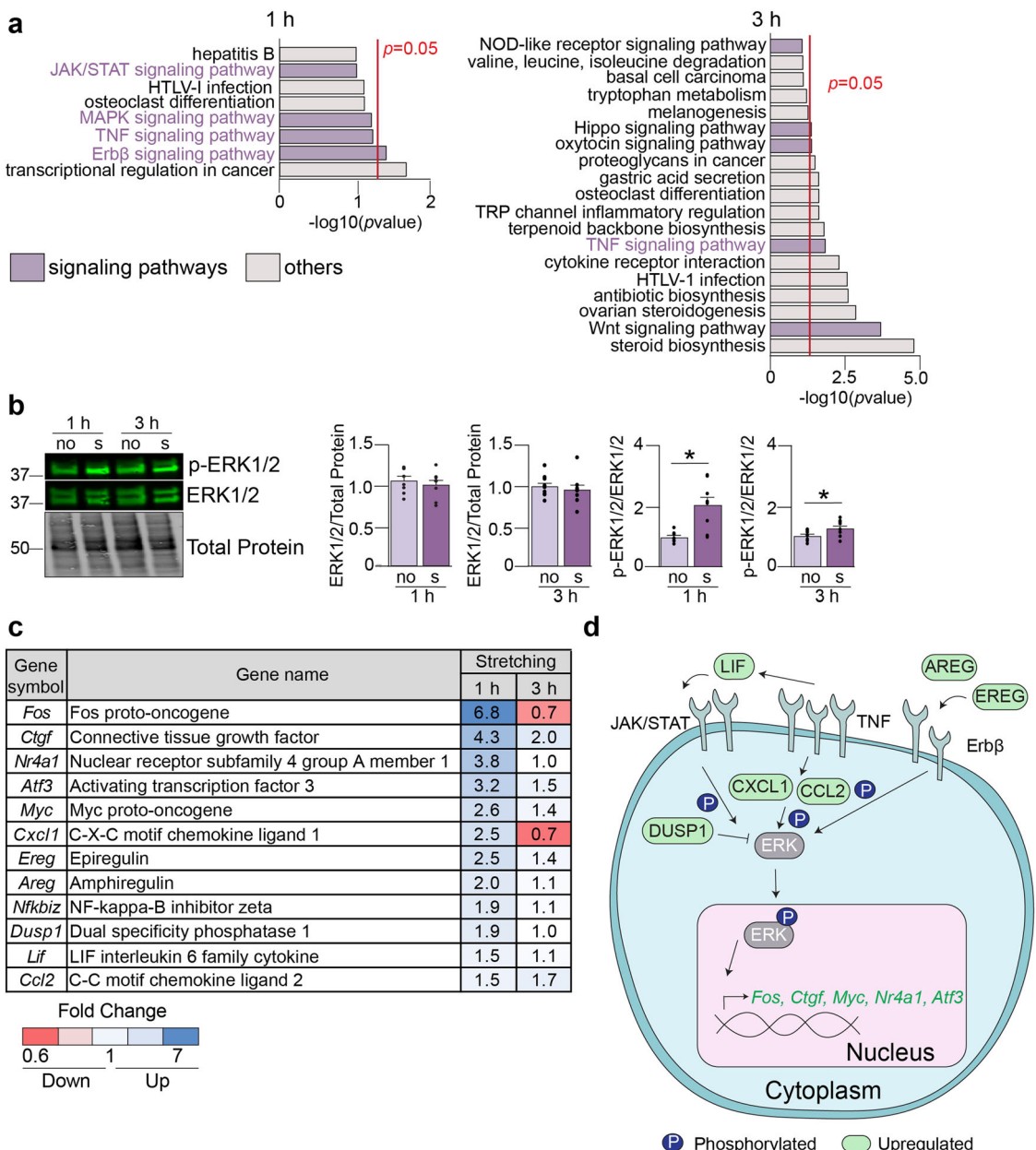

**Fig. 2 Stretching induces activation of the MAPK pathway. a** Pathway analysis of the differentially expressed genes in response to stretching in differentiated cells was performed using DAVID (Database for Annotation, Visualization, and Integrated Discovery). Red lines indicate p-values<0.05. **b** Differentiated muscle cells were stretched for 1 h or 3 h. Western blot assays were performed and quantified by densitometry to examine total ERK1/2 and phosphorylated ERK1/2 (p-ERK1/2) levels. Results are shown as mean ± SEM, N = 3, *p < 0.05, Welch's T-test. **c** mRNA expression changes of various genes encoding proteins involved in signaling pathways linked to ERK1/2 using RNA-seq data (differentiated cells). The fold change was defined as the ratio between stretched and non-stretched samples. Values above 1 indicate upregulated genes and values below 1 indicate downregulated genes. **d** Cartoon depicting the genes activated by cell stretching to illustrate how the MAPK pathway may be activated. no: non-stretched samples. s: stretched samples.

spliced increased over time while the proportion of retained introns decreased (except for the transition from 3 h to 6 h in differentiated cells) (Fig. 5b). We focused the rest of our analysis on cassette exons since these were the most prevalent type of splicing event.

We next interrogated if stretching induced changes in the distribution and mode of cassette exon length. There was more variation in the mode for the myoblasts than for the differentiated cells. However, across all three timepoints, there was a similar distribution of cassette exon length both in myoblasts and differentiated cells (Fig. 5c) suggesting that cassette exon length is not sensitive to stretching.

Cassette exons that are divisible by three can add a peptide onto the protein sequence or have the potential to add a stop codon. Whereas cassette exons that are indivisible by three can interrupt the reading frame often resulting in a truncated protein or a protein with a different C-terminus. We thus determined the proportion of exons that were sensitive to stretching and divisible or indivisible by three. In myoblasts, 48% of cassette exons that responded to 1 h of stretching were divisible by three, 61% and 50% of the exons that responded to 3 h or 6 h of stretching were also divisible by three, respectively (Fig. 5d, top). This indicates that myoblasts may exhibit slightly less truncated or potentially frameshifted proteins after 3 h of stretching. Conversely, in

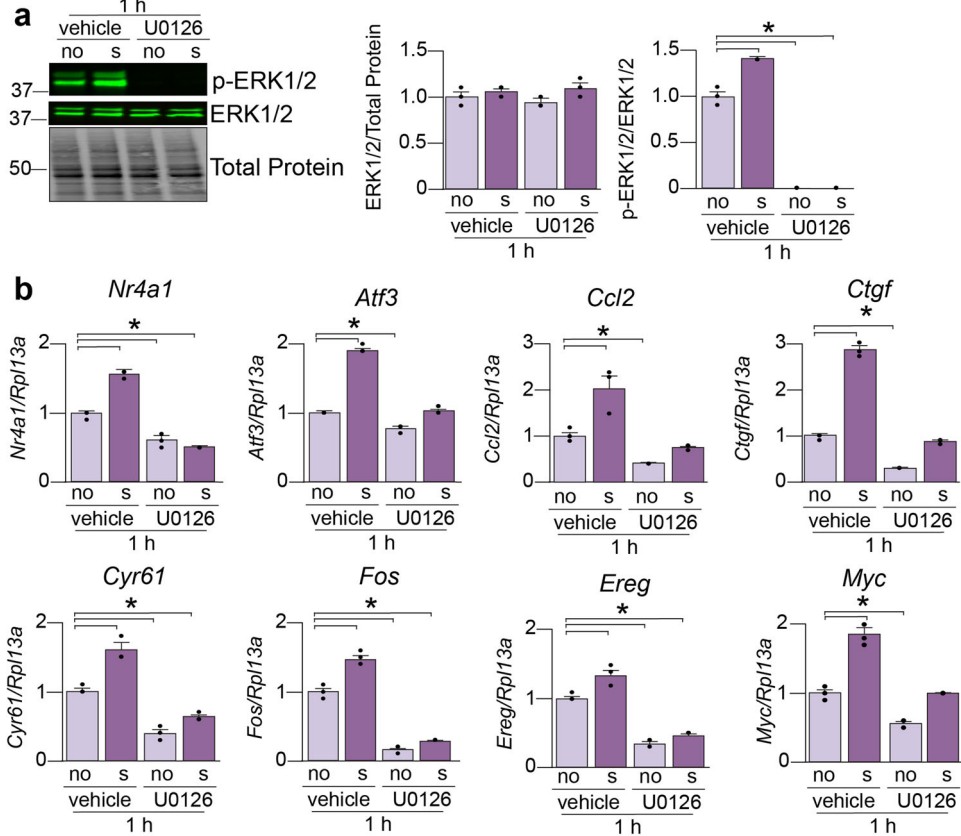

**Fig. 3 Inhibition of ERK1/2 phosphorylation impacts the response of mechanosensitive genes. a** Differentiated muscle cells were treated with the ERK1/2 phosphorylation inhibitor U0126 and stretched for 1 h. Western blot assays were performed and quantified by densitometry to examine total ERK1/2 and phosphorylated ERK1/2 (p-ERK1/2) levels. **b** qPCR assays were performed and quantified to examine the mRNA expression changes of eight mechanosensitive genes. Results are shown as mean ± SEM, $N = 3$. *$p < 0.05$, one-way ANOVA with Dunnett's *post hoc* multiple comparisons test.

differentiated cells all three timepoints had a similar proportion of cassette exons that were sensitive to stretching and divisible by three: 48% (1 h), 45% (3 h), and 43% (6 h) (Fig. 5d, *bottom*).

We then asked whether stretching induced more skipping or more inclusion of the alternatively spliced regions. In myoblasts, the proportion of events with $\Delta PSI > 10$ (stretching induces inclusion) increases over stretching time (Supplementary Fig. 8a, *left*). In differentiated cells, we observed a more drastic trend in the same direction (Supplementary Fig. 8a, *right*). Since we focused most of our analysis on cassette exons we also determined if stretching induced more skipping or inclusion specifically of cassette exons. In myoblasts, the proportion of cassette exons with $\Delta PSI < -10$ (stretching induces exclusion) increases slightly over stretching time (Fig. 5e, *top*). In differentiated cells, we observed an opposite trend with longer stretching times inducing more inclusion of cassette exons (Fig. 5e, *bottom*).

**Stretching promotes exon inclusion in genes encoding proteins involved in transcription, DNA damage, phosphorylation and degradation.** We next performed GO analysis to determine whether genes containing alternatively spliced cassette exons regulated by cell stretching were functionally related. In myoblasts, categories related to transcriptional regulation (1 h, 3 h, 6 h) and DNA damage, and phosphorylation (3 h and 6 h) were enriched (Supplementary Fig. 8b). In differentiated cells, when we analyzed the alternatively spliced genes upon stretching, we observed an enrichment in categories related to transcription regulation (1 h and 6 h), phosphorylation (1 h, 3 h, 6 h), and DNA damage (6 h) (Supplementary Fig. 8c). Genes containing cassette

exons that responded to stretching at all timepoints in myoblasts and for 6 h in differentiated cells were enriched for DNA damage categories. This suggests that stretching differentiated cells for long periods of time may lead to damage.

**Some genes are responsive to stretching regardless of differentiation context.** Myoblasts and differentiated cells experience distinct mechanical environments, but we hypothesized that some genes may be mechanosensitive regardless of differentiation status. To examine this, we overlapped the mechanosensitive genes both at the transcriptional (gene expression) and splicing levels between myoblasts and differentiated cells at each timepoint after stretching. We found that after 1 h of stretching, myoblasts and differentiated cells shared numerous mechanosensitive genes (57 genes); this overlap decreased as cells were stretched longer (37 genes for 3 h of stretching and 43 genes for 6 h of stretching) (Supplementary Fig. 9a). Overall, there were few genes that were alternatively spliced that overlapped between myoblasts and differentiated cells at any timepoint (12-19 genes). Further examination of the mechanosensitive genes in both myoblasts and differentiated cells after 1 h of stretching revealed that several of them were involved in the MAPK pathway (Supplementary Fig. 9b, *green*). *Nr4a1, Atf3, Ctgf*, and *Fos* were confirmed to respond to ERK1/2 phosphorylation inhibition indicating that indeed ERK1/2 is involved in regulating their response to stretching (Fig. 3b).

Previous work has demonstrated that splicing changes result in expression of different protein isoforms as opposed to variation in alterations of overall gene expression levels[37–39]. To examine this,

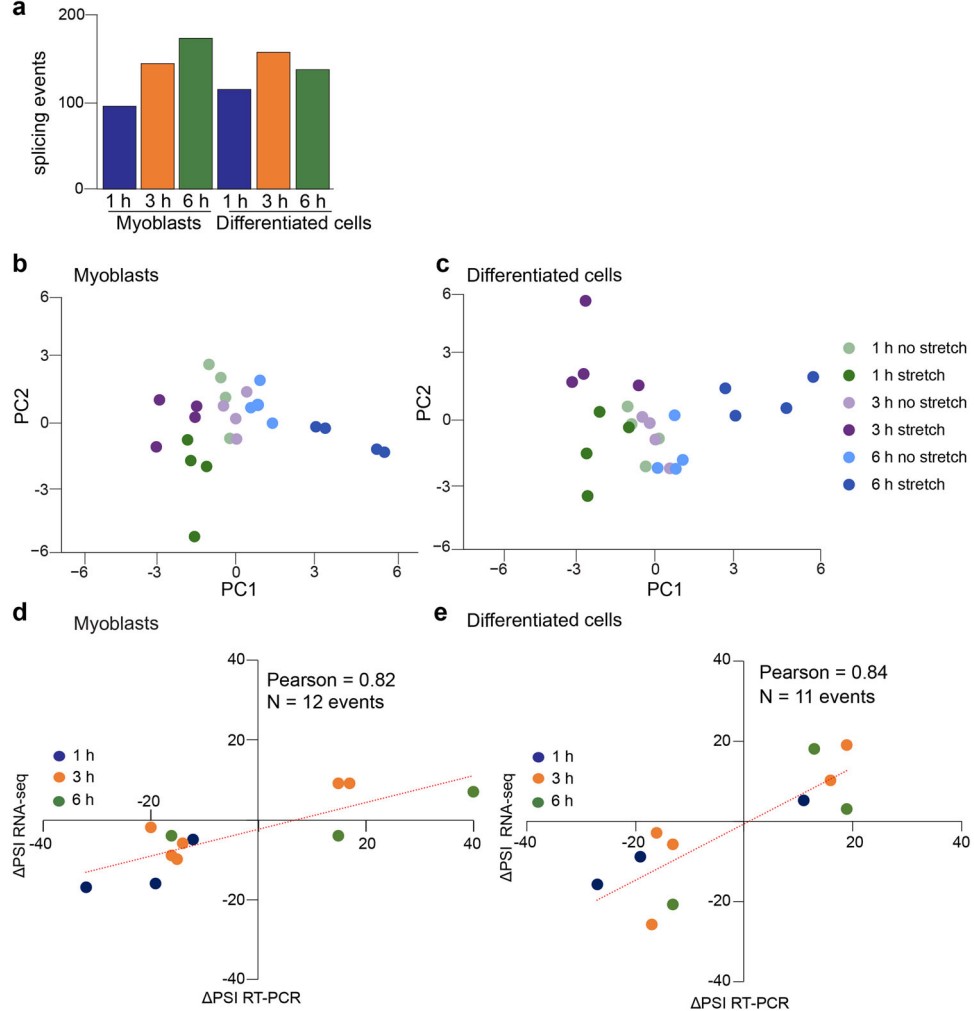

**Fig. 4 Alternative splicing transitions occur in response to stretching and change over time. a** Number of alternative splicing events after 1 h, 3 h, and 6 h of stretching of myoblasts (*left*) and differentiated cells (*right*) identified by RNA-seq experiments. **b**, **c** Principal component analysis (PCA) plots of the alternative splicing changes in response to stretching in myoblasts and differentiated cells. **d**, **e**. Correlation plots of ΔPSI values obtained from RNA-seq studies *versus* ΔPSI values obtained from RT-PCR experiments in myoblasts and differentiated cells. N = 4 in myoblasts and N = 4-5 in differentiated cells. RT-PCR gels producing the data included in the correlation plots are shown in Supplementary Fig. 6 (myoblasts) and Supplementary Fig. 7 (differentiated cells). Alternative regions were considered differentially spliced if | ΔPSI | > 10 between stretched and non-stretched samples. The ΔPSI was defined as the difference between the PSI in stretched samples and the PSI in the non-stretched controls. PSI percent spliced in.

we overlapped the mechanosensitive genes at the transcriptional (gene expression) level with the genes changing their alternative splicing patterns in response to stretching at each timepoint (Supplementary Fig. 10). We found that there was a minimal overlap between the mechanosensitive genes with altered transcription and those changing their splicing patterns. Therefore, transcriptional changes and splicing variation due to cell stretching are involved in distinct cellular processes and are independent of each other.

**A portion of mechanosensitive genes are SR protein targets.** Since longer periods of stretching induced more exon inclusion in differentiated cells we hypothesized that changes in expression, translation, or the activity of specific splicing factors might be involved in the promotion of exon inclusion. To evaluate this, we determined the effect of stretching differentiated muscle cells on the mRNA expression levels of RBPs using our RNA-seq data. Globally, we observed that the group of RBPs exhibiting more pronounced changes were the serine and arginine-rich (SR) proteins. In Eukaryotes, there are 12 SR proteins (named SRSF1

to SRSF12) that regulate alternative splicing and gene expression[40]. These proteins have numerous functions in the cell but are most notably involved in the splicing reaction[41,42].

In our studies, we observed that SRSF2, SRSF5, and SRSF6 exhibited strong mRNA changes after stretching (Fig. 6a). In contrast, other RBPs widely expressed in muscle cells such as the CUGBP Elav-like family member (CELF) proteins, muscleblind like splicing regulator (MBNL) proteins, heterogeneous nuclear ribonucleoparticles (HNRNP) proteins, polypyrimidine tract binding protein 1 (PTBP1), quaking (QKI), and RNA binding fox-1 homolog 2 (RBFOX2) responded minimally if at all to cell stretching (Fig. 6a).

SR proteins are activated by phosphorylation[40] and SR protein phosphorylation could likely contribute to mechanotransduction; however, to our knowledge, no studies have tested this hypothesis. We thus investigated whether any of the genes that responded to stretching (transcriptionally or at the splicing level) were known targets of the SR family proteins. To examine this, we used a previous RNA-seq study that identified the genes and splicing events responsive to overexpression of SRSF2, SRSF4, SRSF6 or SRSF9 in MCF-10A cells[43]. We determined the overlap

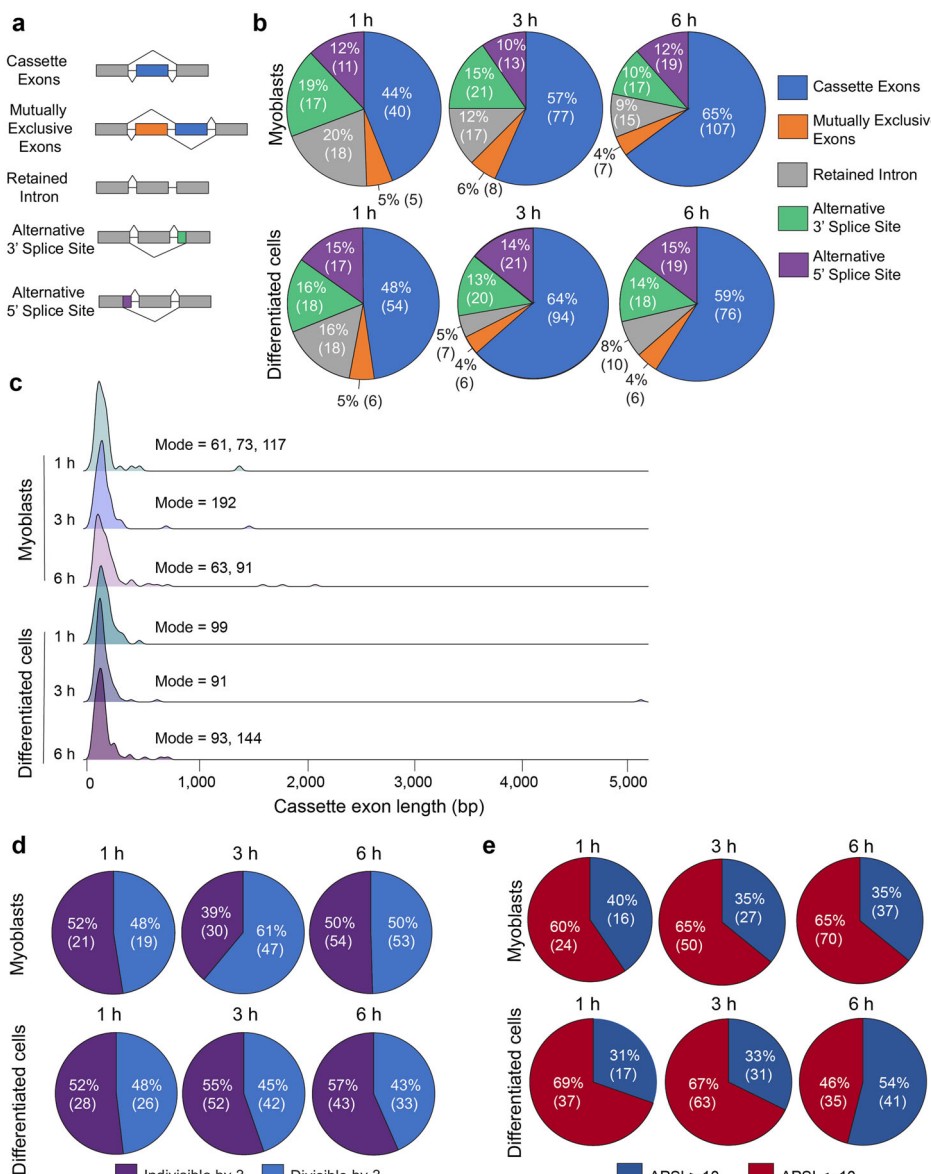

**Fig. 5 Alternative splicing changes in response to stretching occur mostly in cassette exons. a** Schematic showing the different type of splicing events. **b** Distribution of the splicing events depending on their type in myoblasts (*top*) and differentiated cells (*bottom*). The numbers between parentheses indicate the number of splicing events. **c** Density plots of cassette exon length in myoblasts (*top*) and differentiated cells (*bottom*). The mode(s) of cassette exon length is shown next to each plot. **d** Proportion of cassette exons that are indivisible by three (purple) or divisible by three (blue) in myoblasts (*top*) and differentiated cells (*bottom*). The numbers between parentheses indicate the number of splicing events. **e** Proportion of splicing events that underwent more inclusion (ΔPSI > 10, blue) or more exclusion (ΔPSI < −10, red) of the alternatively spliced region upon stretching. The numbers between parentheses indicate the number of splicing events. The ΔPSI was defined as the difference between the PSI in stretched samples and the PSI in the non-stretched controls. PSI percent spliced in.

between the genes that responded to SR protein overexpression (at the transcriptional and splicing levels) and those that were mechanosensitive in our differentiated muscle cells (all time points). Approximately 18% of the genes that responded to stretching overlapped with the genes that responded to SRSF4 overexpression while the percentages were lower for other SR proteins (Fig. 6b). Interestingly, 20% and 21% of stretch-responsive genes at the splicing levels (cassette exons) overlapped with those that responded to SRSF4 or SRSF6 overexpression, respectively (Fig. 6c). Overall, this suggests that stretching cells induces both transcriptional and posttranscriptional changes that may be regulated by SRSF4 in our system.

**SRSF4 is a mechanosensitive RBP.** Activation of SR proteins occurs by phosphorylation, thus, we hypothesized that stretching may induce phosphorylation changes in the SR proteins. We explored some of the SR proteins and investigated their phosphorylation status in differentiated muscle cells. As we described above, splicing changes that responded to 6 h of stretching occurred in genes encoding proteins involved in the DNA damage response (Supplementary Fig. 8c) which suggests that longer periods of stretching could induce a stress response and may be too long of a stimulus for our cells. Further, *Ctgf* and *Cyr61* did not exhibit significant upregulation after 6 h in differentiated cells indicating that the cells became adapted to the

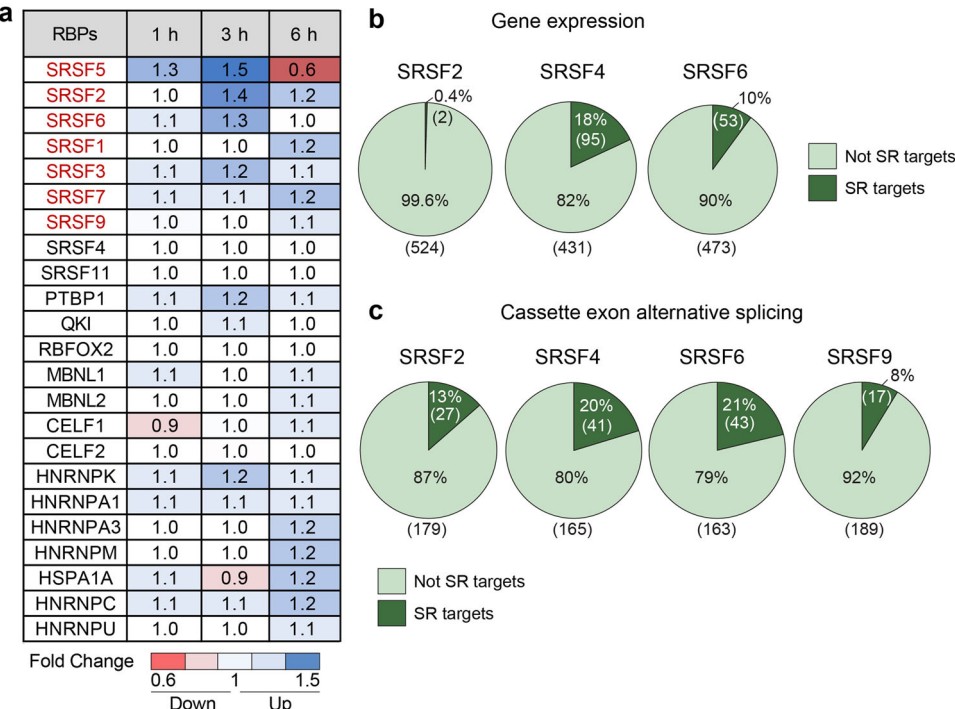

**Fig. 6 A portion of the genes that are mechanosensitive are targets of the SR proteins. a** Gene expression changes of various RNA-binding proteins (RBPs) upon cell stretching of differentiated cells. The fold change was defined as the ratio between stretched and non-stretched samples. Values above 1 indicate upregulated genes and values below 1 indicate downregulated genes. **b** Pie charts exhibiting the overlap between genes that respond to SR protein overexpression[43] at the transcriptional level[43] and genes that respond to cell stretching. The numbers between parentheses indicate the number of genes. **c** Pie charts showing the overlap between genes with alternative cassette exons that respond to SR protein overexpression[43] and genes with alternative cassette exons that respond to cell stretching (splicing level). The numbers between parentheses indicate the number of genes.

stretch stimulus. Thus, we opted to analyze only the 1 h and 3 h timepoints.

We evaluated total protein levels of some SR proteins and their degree of phosphorylation by western blotting. Interestingly, we did observe differences in the levels of phosphorylated SRSF4 between stretched and non-stretched cells (Fig. 7a) even though *Srsf4* mRNA expression did not respond to stretching (Fig. 6a). After 1 h of stretching, SRSF4 total protein was slightly upregulated, but its phosphorylation level was significantly decreased (Fig. 7a). SRSF4 total protein was slightly upregulated in response to 3 h of stretching and this was accompanied by a corresponding increase in its phosphorylation (Fig. 7a). We observed no change in SRSF5 total protein or phosphorylation after stretching (Fig. 7b) even though stretching induced alteration in its mRNA levels. *Srsf6* mRNA expression and SRSF6 total protein were slightly upregulated after 3 h of stretching and this was accompanied by a significant increase in its phosphorylation levels (Fig. 7c).

Thus, SRSF4 phosphorylation appears to be mechanosensitive after 1 h of stretching whereas a longer stretching period (3 h) led to an increase in SRSF4 total protein with a concomitant increase in its phosphorylation. We were interested in determining the potential transcriptional and posttranscriptional programs that SRSF4 controlled when cells were stretched. To examine this, we analyzed the overlapped genes and splicing events that responded to SRSF4 overexpression[43] and cell stretching (1 h and 3 h) (Fig. 7d). We observed that several of those overlapping genes encode proteins that are connected to the MAPK signaling cascade (Fig. 7d). This evidence led us to propose that in muscle cells SRSF4 might act as a mechanosensitive RBP that helps control transcriptional and posttranscriptional programs in response to stretch through the MAPK pathway.

## Discussion

**Muscle cell differentiation and mechanical stretching**. Various groups have stretched muscle cells for varying stretching amplitudes and diverse time intervals reviewed in[25]. Those studies have presented conflicting conclusions with some stating that stretching induces muscle cell proliferation and others proposing that stretching promotes muscle cell differentiation[25]. We observed both upregulation and downregulation of specific differentiation transcription factors in myoblasts. Myogenin is a key driver of myogenesis and was downregulated after stretching. The FOS proto-oncogene is a negative regulator of myogenin and was strongly upregulated after 1 h of cell stretching. This upregulation suggests that FOS downregulates myogenin expression to prevent muscle cell differentiation. Conversely, the transcription factor early growth response 1 (EGR1) is a positive regulator of myogenin[44] and *Egr1* mRNA was upregulated after stretching in our study. Since *Egr1* mRNA is upregulated upon stretching without a corresponding increase in myogenin expression, this suggests that FOS downregulates myogenin expression to prevent myoblast differentiation in response to cell stretching.

Previous studies have established heterogeneity of myoblast cells which has been hypothesized to contribute to their varying responses to mechanical stimulus[25,45]. Our data are consistent with this hypothesis since we observed changes in both proliferation- and differentiation-promoting transcription factors which indicates that the heterogenous myoblasts may generate a complex response to mechanical stretch. When skeletal muscles undergo a mechanical stimulus regeneration occurs more effectively by activation of the satellite cells[46]. Thus, the fact that cell heterogeneity affects the ability of myoblasts to respond to stretching could have implications on the capacity of muscle tissue to regenerate[47].

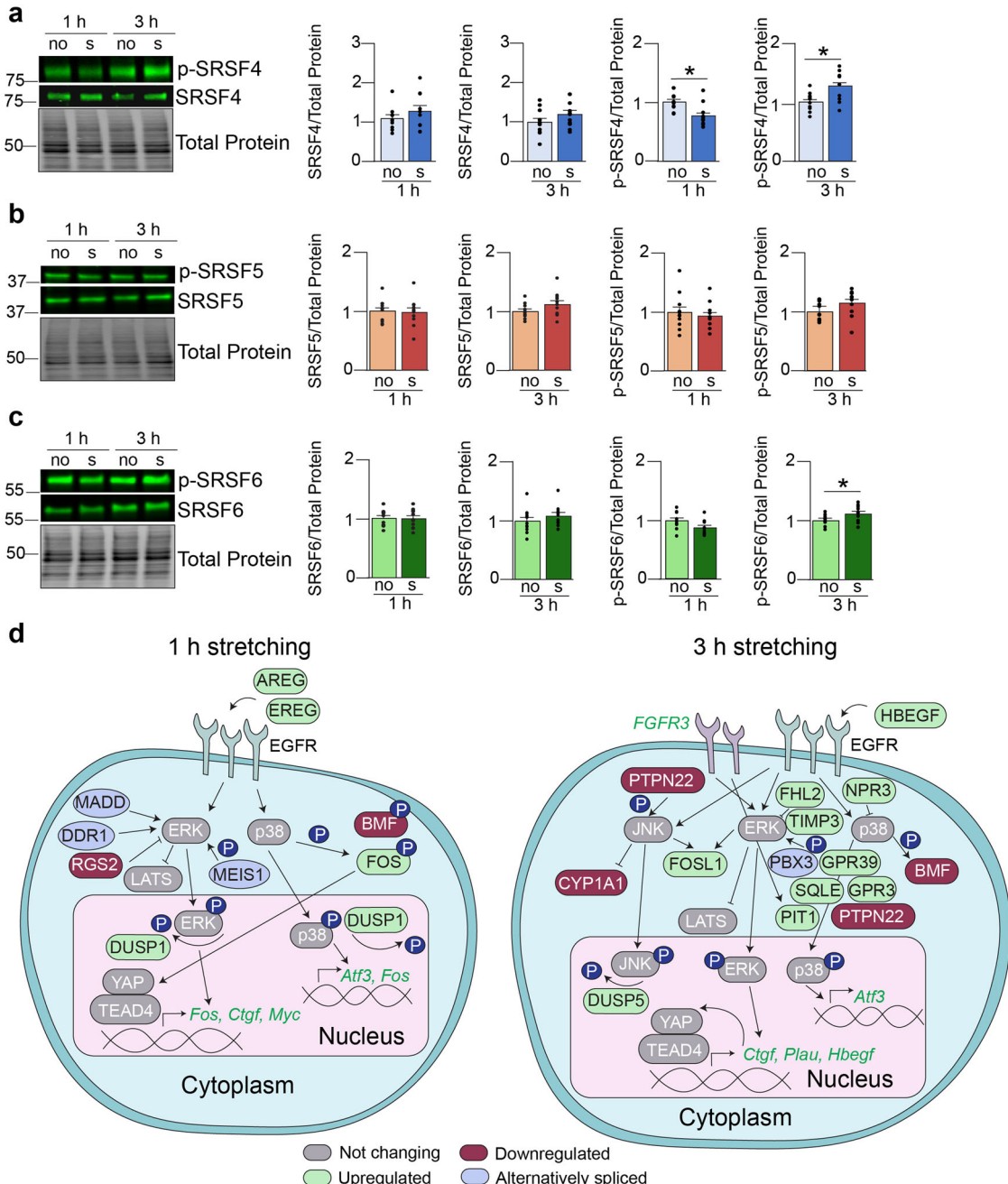

**Fig. 7 SRSF4 is a mechanosensitive RBP.** Differentiated muscle cells were stretched for 1 h or 3 h and western blot assays were performed and quantified by densitometry to examine total and phosphorylated levels of SRSF4 (**a**), SRSF5 (**b**), and SRSF6 (**c**). no: non-stretched samples. s: stretched samples. **d** Some overlapping genes and splicing events from the SRSF4 pie charts shown in Fig. 6c were depicted in cartoons for 1 h and 3 h stretching time points showing how mechanosensitive targets and those regulated by SRSF4 might play a role in the MAPK signaling cascade. ATF3 activating transcription factor 3, AREG amphiregulin, BMF Bcl2 modifying factor, CYR61 cysteine-rich angiogenic inducer 61, CYP1A1 cytochrome p450 family 1 subfamily A member 1, CTGF connective tissue growth factor, DDR1 discoidin domain receptor tyrosine kinase 1, DUSP1 dual specificity phosphatase 1, DUSP5 dual specificity phosphatase 5, EGFR epidermal growth factor receptor, ERK extracellular signal-regulated kinase, EREG epiregulin, FGFR3 fibroblast growth factor receptor 3, FHL2 four and a half LIM domains 2, FOS Fos proto-oncogene AP-1 transcription factor subunit, FOSL1 FOS like 1 AP-1 transcription factor subunit, GPR3 G-protein coupled receptor 3, GPR39 G-protein coupled receptor 39, HBEGF heparin binding EGF like growth factor, LATS large tumor suppressor kinase 1, JNK c-Jun N-terminal kinase, MADD MAP kinase activating death domain, MEIS meis homeobox 1, MYC MYC proto-oncogene, NPR3 natriuretic peptide receptor 3, PBX3 PBX homeobox 3, PIT1 pituitary transcript factor 1, PLAU plasminogen activator urokinase, PTPN22 protein tyrosine phosphatase non-receptor type 22, P38 p38 mitogen-activated protein kinase, RGS2 regulator of G-protein signaling 2, SQLE squalene epoxidase, TEAD4 TEA-domain transcription factor 4, TIMP3 TIMP metallopeptidase inhibitor 3, YAP yes-associated protein. Results are shown as mean ± SEM, $N = 4$ (**a**), $N = 4$ (**b**), $N = 4$ (**c**) *$p < 0.05$, Welch's *T*-test.

Short-term stretching of differentiated cells induced the upregulation of muscle differentiation genes and some proliferation-associated genes. Proteins involved in cellular proliferation and differentiation often overlap because there is a delicate balance of transcriptional programs during myogenesis that drives cells to either proliferate or differentiate[48]. Interestingly, at each stretching time point differentiated cells exhibited mRNA changes in distinct groups of functionally related genes as opposed to increased time under stretch driving a larger magnitude of change. This indicates that muscle cells undergo genetic reprogramming as they become adapted to mechanical stimulus. This is consistent with previous studies where repeated stretch exposure of skeletal muscle fibers alters fiber type by changing gene expression programs as an adaptive mechanism in response to mechanical stimuli[49]. Adaptation of muscle cells to stretching occurs via various signaling pathways[50] which might explain in part why there are more unique gene expression changes after 1 h of stretching in differentiated cells compared to 3 h and 6 h of stretching which overall share more similar transcriptional transitions.

We chose to not directly compare myoblasts to differentiated cells because of their unique mechanical contexts, however, it was interesting that some mechanosensitive genes overlapped between the individual timepoints in both myoblasts and differentiated cells. This suggests that some mechanosensitive genes at the transcriptional and splicing level respond similarly to mechanical force regardless of differentiation status and could be strong candidates for future experiments investigating mechanotransduction in skeletal muscle.

**Stretching activates the MAPK pathway**. Although the MAPK pathway is known to be mechanosensitive in mouse and rat systems[29–31], the molecular mechanisms governing this activation and its functional implications are still unclear. We observed that several of the upregulated genes after stretching were downstream targets of the MAPK cascade and others were upstream activators of MAPK indicating a potential positive feedback loop to promote consistent activation of the pathway. Further, some of the mechanosensitive genes after 1 h of stretching (myoblasts and differentiated cells) are activated by the MAPK pathway. *Fos* is an early response gene that is highly upregulated in response to exercise and involved in a wide variety of cellular processes including transcription[51]. *Ctgf* is another exercise early response gene and CTGF is recruited to muscle fibers likely to promote remodeling of matrix proteins[27]. The strong activation of these two genes after 1 h of stretching in both myoblasts and differentiated cells suggests a role of the MAPK pathway in regulating the stretch response through transcriptional programs. Interestingly, several MAPK targets exhibited an ablated response to cell stretching when ERK1/2 phosphorylation was inhibited indicating that MAPK is indeed involved in transcriptional modulation.

Previous studies have hypothesized that MAPK signaling interfacing with other splicing factors is the key to understanding how alternative splicing responds to mechanical forces[29], but this has not been proven. A recent study conducted in *C. elegans* demonstrated that loss of the splicing factor MBNL1 resulted in decreased activity of MAPK which affected downstream splicing and gene expression programs[52]. In our study, the activation of ERK1/2 upon stretching and the subsequent effects on transcriptional targets suggests that the MAPK pathway could be interfacing with other splicing factors to cause those changes.

**Exon inclusion after cell stretching and potential roles for SR proteins**. Stretching cells for 1 h as well as 3 h led to an increase in the total number of splicing events in differentiated muscle cells

as well as a shift to more exon inclusion over exclusion. SR proteins are well-known promoters of exon inclusion[53,54]. Our RNA-seq studies revealed that mRNA expression levels of the SR proteins respond to stretch more than other RBP families. Once phosphorylated, SR proteins regulate a wide variety of cellular processes including splicing activation, splicing repression, transcription, translation, and mRNA export[40]. The mechanosensitive cassette exons that we found are present in genes that encode proteins involved in transcription and phosphorylation which are two of the main functions of the SR proteins. This lends credence to the idea that splicing events responding to stretch could be involved in similar functions as SR proteins and be regulated by them.

**SRSF4 is a mechanosensitive RBP**. Our data suggests that SRSF4 is a mechanosensitive RBP since stretching alters its phosphorylation levels which could in turn influence downstream transcriptional and posttranscriptional targets. Indeed, there was a strong overlap between the genes that respond to stretching (transcriptionally and at the splicing level, our study) with those that are known SRSF4 targets[43]. Since skeletal muscle cells must integrate mechanical signals, the phosphorylation changes of SRSF4 upon stretching affirms the idea that it is possible to have an SR protein-modification code that responds to mechanical forces and impacts mechanotransduction[40].

Players of the MAPK pathway interact with kinases upstream of some SR proteins and their targets[42,55,56]. Interestingly, several of the genes and splicing events that responded to SRSF4 overexpression and cell stretching are within the MAPK pathway. This suggests a role of SRSF4 interfacing with the MAPK pathway through mechanical stretch-induced alteration of its transcriptional and posttranscriptional targets.

**Physiological implications for health and disease**. Alternative splicing programs are misregulated in a wide variety of muscular diseases[57–59]. Numerous studies have linked splicing alterations to specific features and physiological defects of skeletal muscle diseases and have identified the responsible RBPs[3,4,57,59]. Mechanosensitive tissues such as muscle and heart exhibit high levels of alternative splicing during development[2,14]. Thus, it is important to understand how mechanotransduction impacts alternative splicing regulation to provide more therapeutic options for striated muscle diseases. Muscle stretching experiments in mice and humans often exhibit heterogeneity making it difficult to parse the molecular mechanisms of stretching[60,61]. Thus, while studies of muscle cells grown in a dish cannot be extrapolated to muscle tissue, there is still much value in examining the fundamental, molecular mechanisms of muscle cells to clearly inform targets for future study in tissue.

SRSF4 plays a role in breast cancers and regulates splicing of some mechanosensitive proteins in striated muscle[43,62,63]. In 8% of breast cancers, SRSF4 is mutated resulting in missplicing of important cancer transformation genes[43]. Furthermore, overexpression of SRSF4 in MCF-10A acinar structures misregulated downstream gene expression and splicing programs establishing SRSF4 as an important regulator[43]. Moreover, depletion of SRSF4 in HeLa cells caused a significant increase in the inclusion of exon 7 of the survival of motor neuron 2 (SMN2) pre-mRNA[62]. This is important because improving SMN2 exon 7 inclusion is a goal of therapeutic strategies to treat patients with spinal muscular atrophy, which is caused by defects in the *SMN1* gene resulting in SMN protein deficiency. SMN2 is almost identical to SMN1 but cannot compensate for SMN1 loss when exon 7 is skipped because a dysfunctional truncated SMN2 protein is produced.

In heart, SRSF4 promoted exon inclusion of the mechanosensitive cardiac troponin T gene[63]. Knockout of SRSF4 in mouse hearts led to ventricular hypertrophy and larger cardiomyocyte area along with alterations in glucocorticoid signaling[64]. In differentiated muscle cells, we observed that mechanosensitive genes after 3 h and 6 h of stretching encode proteins involved in steroid metabolism which is interesting considering the role of SRSF4 in steroid signaling in heart[64]. Further, mouse hearts lacking SRSF4 did not exhibit extensive splicing changes, but displayed increased SRSF6 protein in the myocardium suggesting that these two SR proteins may regulate similar targets in cardiac tissue[64]. Our study demonstrated an overlap between genes undergoing posttranscriptional transitions after stretching with the targets of SRSF4 or SRSF6 overexpression which suggests that these two RBPs may regulate similar splicing events. In addition, we observed a significant increase in phosphorylated SRSF6 after 3 h of cell stretching similar to the increase in phosphorylated SRSF4. Overall, these studies and our data support a potentially important role of SRSF4 in the regulation of expression and splicing of specific genes, some of which are mechanosensitive.

Muscular diseases result in radical alterations of both splicing and physical properties of muscle cells. There are still numerous muscular pathologies with unknown causes. To the best of our knowledge, this is the first study that investigated the global transcriptional and posttranscriptional changes due to mechanical stretching with an unbiased RNA-seq approach in skeletal muscle cells. We discovered that SRSF4 phosphorylation is mechanosensitive and no other RBPs have been described to respond to stretching in skeletal muscle cells. SRSF4 may play a role in the molecular crosstalk between mechanotransduction and alternative splicing networks in muscle by interacting with the MAPK signaling pathway. This could be an important step to understanding the connection between the simultaneous molecular and mechanical alterations that occur in muscular diseases. SRSF4 will be an attractive target to study in the future because of its already established impact on transcriptional and posttranscriptional regulation and potential new role in mechanotransduction through the MAPK cascade.

## Method

**Cell culture.** C2C12 mouse myoblast cells were purchased from ATCC (CRL-1722) and verified to be negative for mycoplasma. Cells were authenticated by profiling known myogenic markers in myoblasts and differentiated cells by real time PCR (qPCR). Myoblasts were cultured at 37 °C under 5% $CO_2$ in Dulbecco's Modified Eagle Medium (DMEM) supplemented with 10% fetal bovine serum (FBS), 2 mM glutamine, 100 units/mL penicillin, and 100 µg/mL streptomycin. Cells were maintained at low confluency (30–40%). To induce differentiation, cells at 90% confluency were washed with phosphate buffered saline (PBS) and cultured in DMEM supplemented with 2% horse serum, 2 mM glutamine, 100 units/mL penicillin, and 100 µg/mL streptomycin.

**Cell stretching.** C2C12 myoblast cells were plated in BioFlex Collagen 1 six well plates (Flexcell International, BF-3001 C) (50,000 cells per well for myoblasts' studies and 200,000 cells per well for differentiated cells' studies) in DMEM supplemented with 10% FBS, 100 units/mL penicillin, 100 µg/mL streptomycin, and 2 mM glutamine. The non-stretched cells were also plated on BioFlex Collagen 1 six well plates and subjected to exactly the same culturing conditions as the stretched cells except the period of stretching. Cells were either differentiated for four days and then stretched using a FX-6000T Tension system (Flexcell International) for 1 h, 3 h, or 6 h (differentiated cells) or stretched for 1 h, 3 h, or 6 h two days following plating (myoblasts). The stretching protocol was as follows: ½ sine, 16% equibiaxial stretch, DC = 50%, and FREQ = 1 Hz. According to the Flexcell technical reports, 16% percent of stretch equates to 0.16 strain and −69.5 pressure (kPa). Both stretched and non-stretched cells were washed with PBS before RNA extraction or preparation of protein lysates.

**RNA extraction.** For RNA-seq experiments, cells were washed with PBS and RNA was extracted using a RNeasy Mini Kit from Qiagen (#74104) including a DNase step (Qiagen, #79254) according to manufacturer's protocols. For the other

experiments, cells were washed with PBS and TRIzol reagent was used to extract total RNA according to the manufacturer's protocol.

**RNA-seq.** RNA samples analyzed by RNA-seq passed the following quality parameters: $DV_{200} > 95$, r28S:18 S > 2.8 and RNA integrated number (RIN) > 8.4. Kapa mRNA stranded method was used for library preparation. All samples were pooled, and the library was first run on an Illumina MiSeq Nano to verify sequencing quality. After quality verification, the library was run on one flow cell over four lanes on a NovaSeq 6000S4 sequencer in a paired end $2 \times 100$ cycle run at the High Throughput Sequencing Core at The University of North Carolina at Chapel Hill.

**Bioinformatics.** Samples were processed using the UNC-CH jUNCtion program which aligned samples to the Ensembl mm10 genome and Ensembl mm10 transcriptome using STAR[65] and quantified expression for downstream analysis using the Salmon program[66]. Mapping rates are detailed in Supplementary Data 1. Gene expression was calculated using DESeq2[67]. Genes were considered differentially expressed if the Benjamini-Hochberg adjusted p-value $p_{adjusted} < 0.05$ and $|\log_2\text{-FoldChange}| \geq 0.58$ (fold change >1.50 for upregulated genes, or fold change < −1.50 for downregulated genes). Significant changes in gene expression for myoblasts are in Supplementary Data 2 and for differentiated cells are in Supplementary Data 3. To determine differential alternative splicing in response to stretching, reads were aligned to the Gencode mm10 genome and Gencode vM20 transcriptome that contained reference chromosomes, scaffolds, assembly patches, and haplotypes. Differential splicing was determined using MISO and datasets were created that contained the MISO summary counts of all samples at each timepoint[35]. Those datasets were then used to compare the stretched samples to the non-stretched controls at each timepoint. Events were filtered by total read depth ≥15 and either inclusion reads ≥5 or exclusion reads ≥5. After filtering, events were considered alternatively spliced if $|\Delta PSI|$ ($|PSI_{stretch} - PSI_{no\ stretch}|$) >10 and the Wilcox p-value (stretch versus no stretch) ≤0.05. Significant changes in alternative splicing for myoblasts are in Supplementary Data 4 and those for differentiated cells are in Supplementary Data 5. In the excel files the ΔPSI values are shown in a scale from 0 to 1 (for inclusion) or 0 to −1 (for exclusion).

**GO.** GO enrichment analysis was performed on genes that changed their mRNA levels or alternative splicing patterns in response to cell stretching using Database for Annotation, Visualization, and Integrated Discovery (DAVID)[68,69]. The differentially expressed genes as defined from the previous section were utilized for GO. The −log10 adjusted p-value for GO analysis of genes changing upon stretching was calculated from the Benjamini-Hochberg adjusted p-value outputted from DAVID. GO analysis with adjusted p-value of 0.25 is a well-accepted threshold for hypothesis generation for GO[70]. The −log10 p-value for pathway analysis of genes changing upon stretching was calculated from the p-value that was outputted from DAVID. The genes (in which the differentially spliced exons resided) as defined from the previous section were utilized for GO. The −log10 ranked p-value from the GO analysis of genes changing at the splicing level (cassette exons) was calculated from the ranked p-values outputted from DAVID.

**cDNA synthesis for qPCRs and RT-PCRs.** The High-Capacity cDNA Reverse Transcription kit (Applied Biosystems, #4368813) was used to reverse transcribe RNA into cDNA with nuclease free $H_2O$ and RNase inhibitor (Applied Biosystems, N8080119). The following program was used for cDNA synthesis: (i) 25 °C for 10 min, (ii) 37 °C for 120 min, (iii) 85 °C for 5 min, (iv) 4 °C pause.

**Alternative splicing assays.** PCR assays were performed using the GoTaq green master mix (Promega, M7123) and mouse primers (0.5 µM) that targeted constitutive exons flanking the alternatively spliced regions (Supplementary Data 6). The following amplification conditions were used: (i) 95 °C for 1 min 15 s, (ii) 28 cycles of 95 °C for 45 s, 57 °C for 45 s, 72 °C for 1 min, (iii) 72 °C for 10 min, (iv) 4 °C pause. The PCR products were separated by electrophoresis using a 6% polyacrylamide gel in TBE buffer (89 mM Tris, 89 mM boric acid, 2.5 mM EDTA, pH 8.3) for 4 h at 140 V. The gels were stained for 10 min in an aqueous solution of 0.4 µg/mL ethidium bromide and visualized using the ChemiDocTM XRS + imaging system (BioRad). Splicing gels were quantified by densitometry using the Image LabTM 6.0.1 software (BioRad).

**qPCR.** 25–100 ng of cDNA was utilized to perform a 20 µL qPCR reaction using the Applied Biosystems TaqMan Fast Advanced Master Mix (Thermo Fisher Scientific, #4444557). A QuantStudio 7 machine was used with the following protocol: (1) 50 °C for 2 min; (2) 95 °C for 20 s; (3) 95 °C for 1 s; (4) 60 °C for 20 s. The cycle threshold values for myoblast and differentiated cells qPCR validation (Supplementary Table 1) were normalized to hydroxymethylbilane synthase (Hmbs) (Mm01143545-m1, amplicon size 81 bp; Thermo Fisher Scientific). The stretched samples were normalized to the non-stretched samples from the same timepoint. The cycle threshold values for qPCRs after U0126 treatment (Supplementary Table 1) were normalized to ribosomal protein L13a (Rpl13a) (Mm01612986_gH,

amplicon size 122 bp; Thermo Fisher Scientific). Samples were normalized to the vehicle, no stretch sample.

**Protein lysate preparation**. Cells were placed on ice immediately after stretching, washed with ice cold PBS with protease (Roche, #11697498001) and phosphatase inhibitors (Sigma Aldrich, #4906845001), and lysed with ice cold RIPA buffer (50 mM Tris, 150 mM NaCl, 1% Triton X-100, 0.5% sodium deoxycholate, 0.1% sodium dodecyl sulfate (SDS), pH 7.5) containing protease and phosphatase inhibitors (Thermo Fisher Scientific, #1861281). Lysates were incubated on ice for 15 min, and then sonicated at 75% amplitude for 3 min (30 s on, 30 s off), incubated on ice for another 15 min and centrifuged at 14,000 r.p.m. for 10 min at 4 °C. Supernatants were transferred to new tubes and stored at −80 °C. Protein concentrations were measured following manufacturer's protocols of the Pierce BCA protein assay kit (Thermo Fisher Scientific, #23225).

**Western blotting**. Equal amounts of protein were diluted in loading buffer (50 mM Tris-HCl pH 6.8, 12.5 mM EDTA, 10% glycerol, 2% SDS, 0.02% bromophenol blue, 360 mM beta-mercaptoethanol). SDS polyacrylamide gel electrophoresis (SDS-PAGE) was used to analyze samples on 4–15% mini-Protean TGX stain-free gels (BioRad, #4568084) utilizing a buffer containing 192 mM glycine, 25 mM Tris base, and 3.5 mM SDS, pH 8.3. Electrophoresis was performed at 90 V for 30 min followed by 150 V for 30 min. Gels were imaged on a ChemiDoc™ XRS + imaging system (BioRad) and proteins were transferred to an Amersham Hybond Low Fluorescence 0.2 µm PVDF membrane (GE Healthcare Life Sciences, #10600022) at 100 V for 1 h using a transfer buffer containing 192 mM glycine, 25 mM Tris, and 20% methanol, pH 8.3. Membranes were imaged using a ChemiDoc Imaging System (BioRad) to estimate total protein. Membranes were blocked for 1 h at room temperature with either 5% nonfat dry milk in Tris-buffered saline containing tween (TBST) (20 mM Tris base, 137 mM NaCl, 0.1% tween 20, pH 7.6) or 1% bovine serum albumin (BSA) in TBST. Membranes were washed with TBST and incubated overnight at 4 °C with the primary antibodies diluted in 1% BSA in TBST as follows: anti-phosphoERK1/2 from Cell Signaling (#4370, 1:2,000), anti-ERK1/2 from Cell Signaling (#4695, 1:2,000), anti-SRSF4 from Millipore Sigma (#06-1367, 1:500), anti-SRSF5 from Millipore Sigma (#06-1365, 1:500), anti-SRSF6 from Bethyl Laboratories (A303-669A-T, 1:1,000), and anti-phosphorylated SR proteins from Sigma Aldrich (MABE-50, 1:750). The next day, membranes were washed in TBST three times (10 min each) and incubated with the secondary anti-mouse Dylight 800 (Thermo Fisher Scientific, SA5-35521) or anti-rabbit Dylight 800 (Thermo Fisher Scientific, SA5-35571) antibodies diluted 1:10,000 in 1% BSA in TBST for 1 h in the darkness at room temperature and rocking. Membranes were imaged with an Odyssey Licor Imager.

**ERK1/2 phosphorylation inhibition**. C2C12 cells were differentiated for four days and then incubated with 20 µM of ERK1/2 phosphorylation inhibitor (Promega, U0126) for 30 min before stretching. Cells not treated with the ERK1/2 phosphorylation inhibitor were treated with dimethyl sulfoxide (DMSO) (vehicle). Cells were stretched for 1 h. Protein and RNA were immediately isolated for subsequent analyses.

**Statistics and reproducibility**. For myoblast RNA-seq, the samples were collected from two independent experiments (two separate samples from each experiment, so four samples in total). For the differentiated cells RNA-seq, the samples were collected from three independent experiments (two separate samples from one experiment, and one sample from the other two independent experiments, so four samples in total). For all the other experiments, at least three independent biological replicates were generated. The type of statistical analysis and number of replicates are indicated in the legend of each figure.

**Reporting summary**. Further information on research design is available in the Nature Research Reporting Summary linked to this article.

## Data availability
The RNA-seq data generated during the current study is available in the GEO repository under accession number GSE190029. The source data for each figure are available at https://doi.org/10.6084/m9.figshare.c.6134205.v2.

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

## Acknowledgements

This work was supported by start-up funds (J.G.) and a Jefferson Pilot Award (J.G.) from The University of North Carolina at Chapel Hill, a National Institutes of Health (NIH) R01 (NIH-NIGMS R01GM130866) (J.G.), a NCTraCs Pilot Grant (550KR181805) from the National Center for Advancing Translational Sciences (NCATS) (J.G.), and a Career Development Award from the American Heart Association (19CDA34660248) (J.G.). E.R.H was supported by a NIH-NIAMS F31 predoctoral fellowship (F31AR077381) and a NIH-NIGMS training award (5T32 GM007092). We gratefully acknowledge the technical support from the High Throughput Sequencing Facility at The University of North Carolina at Chapel Hill, which is supported by the University Cancer Research Fund, Comprehensive Cancer Center Core Support grant (P30-CA016086), and UNC Center for Mental Health and Susceptibility grant (P30-ES010126). M.C. was supported by a NIH-NIAMS F31 predoctoral fellowship (1F31HL145983-01). J.M.T. was supported by a NIH-NHLBI R01 (1R01HL142879). The content is solely the responsibility of the authors and does not necessarily represent the official views of the National Institutes of Health or any other funding source. We thank the Bioinformatics Core at The University of North Carolina at Chapel Hill for their continued help during this project and their jUNCtion program for analyzing data. We acknowledge the support of the Genetics and Molecular Biology Curriculum (GMB) at The University of North Carolina at Chapel Hill. We especially thank Dr. Keith Burridge for numerous valuable discussions regarding this project.

## Author contributions

E.R.H. performed all experimental work and analyzed RNA-seq data. R.E.B. aided in some stretching experiments, performed some western blot and qPCR assays for SR proteins, and helped with western blotting experiments to address reviewer comments. Y-H.T. contributed to MISO data analysis and interpretation. J.D. aided in qPCR analysis. A.R.C. provided some code for MISO analysis. A.M.B. provided stretching protocols in the initial stages of the project. M.C. provided training and support during the utilization of the Flexcell equipment. J.M.T. aided in idea generation for the manuscript. J.S.P. trained and supervised E.R.H in bioinformatics and helped with RNA-seq data interpretation. J.G. designed and supervised the study. E.R.H and J.G. wrote the manuscript draft. All authors contributed to the final manuscript and agreed with it.

## Competing interests

The authors declare no competing interests.
