## [Peer Review File · Communications Biology]

Reviewers' comments:

Reviewer #1 (Remarks to the Author):

This paper investigates how stretching affects gene expression and splicing in myoblasts vs differentiated skeletal muscle cells in vitro. To address this question, the authors performed bulk RNA seq to determine transcriptomic changes in these cells stretched for 1h, 3h or 6h. Authors showed that stretching muscle cells altered gene expression and splicing patterns. They identified genes within the MAPK signaling affected in response to stretch and provided data for activation of ERK kinases in response to stretch. They found that splicing regular SRSF4 is modulated in response to stretch. They concluded that changes in splicing and gene expression in response to stretch partially overlap with splicing and gene expression changes previously identified in SRSF4 overexpressing cells.

This paper addresses an intriguing question related to muscle biology. However, the study is observational and correlative. The conclusions are mostly based on gene ontology analysis of genes without strong support from experimental data. Some important controls are also missing related to stretching and differentiation index of the cells. The link between MAPK signaling and SRSF4 in response to stretching is weak. Moreover, the link between SRSF4 and splicing changes in muscle cells in response to stretch needs to be addressed experimentally. Statements regarding the SRSF4 being mechanosensitive needs to be supported with additional data. Overall, paper has many flaws and needs a major revision to support their conclusions.

Major concerns:

- 1) The role of MAPK in gene expression and splicing changes in response to stretch needs to be addressed.
- 2) Do cells respond to stretch similarly if MAPK or ERK kinases are modulated (inhibited or activated)?
- 3) Does the differentiation status of the cell impact MAPK/ERK signaling?
- 4) Did the authors try to inhibit MAPK or ERK pathway? What happens to myoblasts vs myotubes that are being stretched in the presence of MAPK or ERK inhibitors or activators? Does this affect muscle contraction, forces generated or any phenotype?
- 5) The authors should provide controls for differentiation status of each sample.
- 6) Controls are needed to provide the stretching status of these cells by measuring the forces generated by myoblasts vs differentiated cells in response to stretch. This is very important because forces generated on myoblasts vs differentiated cells can be different even though stretching conditions are the same. The comparisons of gene expression and splicing changes among these groups in response to stretch become hazy.
- 7) To accurately and meaningfully compare these datasets, there should be a normalization standard for differentiation and stretching "force generated" status of these cells to be able to conclude that these changes are mechanosensitive. The effectiveness of stretch in each sample within each group needs to be validated by checking markers for stretch or some other means. There is variation among samples even within same group.
- 8) It is hard to distinguish if mRNA and splicing changes are due to stretching. Controls for muscle differentiation are missing. Did same level of differentiation occur in each sample with or without stretching?. And it would be nice to see the muscle vs differentiated cells in brightfield image in addition to expression of molecular markers for differentiation.
- 9) Authors mention that results from cell stretching is controversial (length and level of stretching and changes on muscle differentiation or proliferation). It is not well justified why authors chose this level and length of stretching in muscle cells.
- 10) What is the rationale of stretching 1h, 3h and 6h? Are these physiological levels? They mention something about stress being induced due to stretching. Do these conditions represent pathological conditions? Authors mention in the intro that the stretching is a part of contraction of skeletal muscle. How much force is generated after 1h, 3h, 6h of stretching on myoblasts vs differentiated myotubes?
- 11) Focusing on a specific target gene relevant to muscle disease or muscle function is necessary to put a perspective on the significance of these findings and to the statement that these genes are

mechanosensitive. The conclusions of this paper are based on GO analysis of gene expression and splicing changes. GO analysis is a good way to find out overall processed affected but there could be other important changes in essential genes (only a few) that do not necessarily form a group in GO analysis. It is important to assess the consequences of these changes on muscle cell physiology.

12) Which genes are stretch sensitive? These should be the genes that are commonly affected in all stretched cells independent of the differentiation status of the cells. I did not see this analysis in the paper. Ideally stretch sensitive genes undergo changes in response to stretch regardless of the cell differentiation status.

13) Myoblasts represent early developmental stages. But same stretching conditions were used for both myoblasts and differentiated cells that represent adult stages. The stretches applied to these cells might not be the same as the muscle tissue at embryo vs adult stages. This should be addressed.

14) It is important that authors overexpress WT or non-phosphorylatable SRSF4 in myoblasts or differentiated cells to examine how global splicing is affected and to test if these cells respond to stretch similarly.

15) To determine direct targets of SRSF4 affected in stretched muscle cells via splicing, it would be better to perform SRSF4-CLIP sequencing analysis or use available SRSF4 CLIP-seq datasets from ENCODE project.

16) How is SRSF4 regulated in response to stretch in muscle cells? This is not thoroughly addressed. SRSF4 levels are increased according to the RNA-seq data. What's the reason for increased SRSF4 mRNA levels?

17) SRSF4 protein phosphorylation is increased. Is phosphorylation of SRSF4 MAPK/ERK dependent?

18) Figure 5: There is unusually high number of out of frame splicing changes. This should be discussed why? Are there any examples of this in muscle diseases or in other conditions?

19) Figure 6A: Authors state that DNA damage is affected based on GO analysis. If DNA damage is present in response to stretch, this should be validated in these cells upon stretch at different time points. GO analysis is not adequate for such a conclusion. This is important because DNA damage can cause apoptosis.

20) Figure 1B: There is no obvious clustering based on time point or stretch vs non-stretched cells

21) Fig 1E & F: Criteria for selecting these genes to show correlation is not explained

22) Fig 2: Based on the log₁₀ value only GO term correlated to data seems to be reg. DNA-templated transcription. The other GO p-values are high to be significant.

23) Fig 4B: PC analysis does not show clustering for splicing datasets

24) Fig 5D- if 60-80 % of CE events are out of frame transition, does this correlate with truncated protein: or is there any evidence of mRNA degradation?

25) Fig6A: Log₁₀ pvalue are too low for any conclusive GO terms

26) Fig8 – data indicate a correlation between the gene expression and splicing but these is not enough data to draw a causative conclusion.

Minor issues:

1) The transition from Figure 1 to Figure 2 is not very smooth. The rationale for analyzing splicing is not well justified in the results section.

2) Figure 5 and 6 can be combined into one figure. The rest of details can be added as a supplemental figure.

3) Figure 1 and 2 can be combined into one figure and the rest of details can be added as a supplemental figure.

4) Comments on splicing changes affecting proteome without proteomic analysis should be toned down.

5) Splicing events are shown as % in pie charts. It is hard to see the exact number of splicing events and mRNA level changes from the graph plots. It is important to add the total number of splicing events and how many total genes undergo splicing changes and mRNA level changes in figure legends.

Reviewer #2 (Remarks to the Author):

In this manuscript, Hinkle et. al. studied the transcriptional changes that occur in response to mechanical loading in both un-differentiated and differentiated muscle cells. They observe changes in gene expression (both upregulated and downregulated) over a range of mechanical loading durations. In particular, the authors observe changes in the MAPK pathway and SR protein phosphorylation. Overall, I find the results interesting and believe this contribution would be of interest to both researchers working in mechanobiology and with muscle cells.

My primary expertise is in mechanobiology, and I defer to the other reviewers for detailed feedback of the RNA sequencing methodology presented by the authors. I have two major points related to the work:

Firstly, I believe the manuscript would benefit from more context describing the mechanical loading parameters and the motivation for choosing these parameters. The authors state that 16% biaxial stretch for 1 h, 3 h and 6 h was used. Would a longer time for stretching cycle change the results in any way? Similarly, biaxial stretch does not seem to be the correct choice for muscle, as muscle fibers are loaded primarily along one axis. This context of the mechanical loading might in-fact have an important impact on the results, given that it is known that cytoskeletal reorganization will respond to the direction of mechanical loading on flexible substrates.

Secondly, I found it difficult to place these findings in a specific physiological context. The authors describe a broad range of possible implications for the results, including development and muscular diseases, which is very interesting, but which of these is best described by the specific mechanical loading conditions that are used in this study? For example, the authors compare no stretch with 1 h, 3 h and 6 h of stretching, but adult muscle presumably is subject to constant cyclic strain (for example in the heart). This again implies that an even longer time for the loading protocol might be more appropriate, and a higher strain magnitude.

Reviewer #3 (Remarks to the Author):

In this work, Hinkle et al have examined the changes in splicing activities and gene expression of cells that have undergone a mechanical stretch. This is a very interesting topic because, for obvious reasons, it is crucial for the proper functioning of all muscle cells in higher organisms. In this work, the authors have presented a comprehensive picture of this phenomenon and have highlighted some specific events, such as identifying SRSF4 as a particular splicing-controlling factor that might be specifically sensitive to this kind of stimuli. Another novelty of the study is represented by the fact that the analysis was performed at two different stages of muscle differentiation, thus providing some insight also from this aspect.

To achieve this aim, the authors have used standard RNAseq and bioinformatics approaches on C2C12 cells. Overall, experiments are uniformly convincing, and the study will certainly be of interest to both splicing and muscle researchers. Nonetheless, some additional clarifications are in order:

1) In Fig.2, it would be interesting to add a table or Venn diagram showing what exactly are the genes that overlap between the myoblasts and the differentiated cells.

2) In Fig.5, the authors show that cassette exons that are divisible by three are more common in differentiated cells over time and suggest that as differentiated cells are stretched longer, there will be more instances of proteins having a defined, internal sequence added to them. However, this does not consider that exons divisible by three may also carry translational stop codons. Can this be analysed in any way?

3) In Figure 6, what is the correlation (if any) between the genes that are alternatively spliced and those that were transcriptionally regulated? Has this been looked at?

4) In Figure 7, what is the overlap if the results of SRSF2, SRSF5, and SRSF6 are pooled together and

compared to the stretched cells? Have the authors tried different combinations to see which pooled results cover the highest proportion of observed changes?

Major Comments:

1. The role of MAPK in gene expression and splicing changes in response to stretch needs to be addressed

We utilized a well-established ERK phosphorylation inhibitor (U0126) and stretched differentiated cells for 1 h. Several known targets of ERK were downregulated in response to ERK inhibition (compare first and third bars in **Figure 3b** and below). For three of the tested genes (*Nr4a1*, *Atf3*, *Ccl2*), we observed that U0126 abolished the response to stretch (first three panels of **Figure 3b**, below). For some other genes (*Ctgf*, *Cyr61*, *Myc*, *Fos*, *Ereg*) the response to stretching was partially abolished in the presence of U0126. Overall, these new data indicate that ERK plays a part in the activation of these genes in response to stretching. We have added these studies in **Figure 3b**.

Figure 3b. ERK phosphorylation inhibitor affects the response of mechanosensitive genes. b. Differentiated muscle cells were treated with an ERK phosphorylation inhibitor (U0126) and stretched for 1 h. qPCR assays were performed and quantified to examine the gene expression changes of eight mechanosensitive genes. Results are shown as mean \pm SEM, $N=3$. $*p<0.05$, one-way ANOVA with Dunnett's *post hoc* multiple comparisons test.

We assayed the exon inclusion of some mechanosensitive genes after ERK phosphorylation inhibition. Both *Hdac7* and *Per1* exhibited less exon inclusion of their alternative exons after 1 h of cell stretching (leftmost bars in **Image 1 for response to reviewers**). We observed that ERK phosphorylation inhibition did not affect the level of inclusion of the alternative exons of *Hdac7* or *Per1* pre-mRNA after 1 h of cell stretching (compare vehicle and U0126 graph bars in **Image 1 for response to reviewers**). This indicates that ERK phosphorylation does not regulate the splicing response of these genes to stretching.

Image 1 for response to reviewers. ERK phosphorylation inhibitor does not affect the alternative splicing of mechanosensitive cassette exons. Differentiated muscle cells were treated with an ERK phosphorylation inhibitor (U0126) and stretched for 1 h. PCR assays were performed and quantified by densitometry to examine the impact of ERK inhibition on alternative splicing after cell stretching. Results are shown as mean \pm SEM, $N=3$. * $p<0.05$, Welch's t-test.

In the manuscript, we observed that some of the genes that are alternatively spliced after cell stretching were part of in the ERK pathway (**Figure 7d**). We did not think that splicing changes were mediated by ERK. Thank you for helping us clarify this. We have made this more clear in the manuscript on **pages 36-37**.

2. **Do cells respond to stretch similarly if MAPK or ERK kinases are modulated (inhibited or activated)? / Did the authors try to inhibit MAPK or ERK pathway? What happens to myoblasts vs myotubes that are being stretched in the presence of MAPK or ERK inhibitors or activators? Does this affect muscle contraction, forces generated or any phenotype?**

We will address these points together. This topic is also related with point #1 addressed above.

Our original **Figure 2** focused on MAPK signaling in differentiated cells, thus, we inhibited ERK phosphorylation (U0126) in differentiated cells and then stretched differentiated cells for 1 h. There was a complete ablation of ERK phosphorylation in both stretched and non-stretched cells after using the inhibitor. We did not observe any morphological differences besides the gene expression changes determined in the above response (see point #1). We have added these western blot assays and their quantifications to the new manuscript (**Figure 3a**).

Figure 3a. ERK phosphorylation inhibitor completely ablates ERK phosphorylation. a. Differentiated muscle cells were treated with an ERK phosphorylation inhibitor (U0126) and stretched for 1 h. Western blot assays were performed and quantified by densitometry to examine total and phosphorylated (p) levels of ERK1/2. Results are shown as mean \pm SEM, $N=3$. * $p<0.05$, one-way ANOVA with Dunnett's *post hoc* multiple comparisons test.

3. Does the differentiation status of the cell impact MAPK/ERK signaling?

We further examined ERK1/2 phosphorylation levels in myoblasts and differentiated cells and observed that it is drastically decreased during C2C12 cell differentiation (**Image 2 for response to reviewers**). In contrast, total ERK1/2 protein expression did not change during C2C12 cell differentiation. In the myoblast state, cells are proliferating and secreting multiple growth factors. Phosphorylation of ERK is highly sensitive to growth factors binding their receptors at the plasma membrane and has been shown to be downregulated upon withdrawal from the cell cycle that occurs during myogenesis (Bennet and Tonks, *Science*, 1997, PMID: 9360925). Thus, it is even more striking that stretching differentiated cells results in such robust ERK1/2 phosphorylation.

Citations:

Bennet and Tonks. *Science*, "Regulation of distinct stages of skeletal muscle differentiation by mitogen-activated protein kinases.", 1997.

Image 2 for response to reviewers. Phosphorylation of ERK1/2 in myoblasts and differentiated cells. Phosphorylation of ERK1/2 (p-ERK1/2) and total levels of ERK1/2 were assayed in undifferentiated myoblasts and differentiated cells by western blotting. Myosin heavy chain 3 (MYH3) was assayed to verify differentiation status of the samples. Results are shown as mean \pm SEM, $N=3$. * $p<0.05$ Welch's T-test. UD: undifferentiated myoblasts. D4: Day 4 of differentiation.

4. The authors should provide controls for differentiation status of each sample

We thank the reviewer for pointing this out to us. We plotted the normalized reads of myoblasts and differentiated cells for various markers of myogenesis. We confirmed that these markers were strongly upregulated in differentiated cells compared to myoblasts. We have added this data as a supplemental figure (**Supplemental Figure 1**) and described it on **page 5**.

SUPPLEMENTAL FIGURE 1

Supplemental Figure 1. Myogenic markers in myoblasts and differentiated cells. Normalized RNA-seq reads were utilized to compare the gene expression of the following myogenic markers in myoblasts and differentiated cells: myogenin (Myog) myosin heavy chain 1 (Myh1), myocyte enhancer factor 2c (Mef2c), and troponin T1 (Tnnt1). Results are shown as mean \pm SEM. no: non-stretched samples. s: stretched samples.

- It is hard to distinguish if mRNA and splicing changes are due to stretching. Controls for muscle differentiation are missing. Did same level of differentiation occur in each sample with or without stretching? And it would be nice to see the muscle vs differentiated cells in brightfield image in addition to expression of molecular markers for differentiation.

At the onset of stretching, the differentiation levels are the same between stretched (s) and non-stretched (no) samples because cells were all cultured together and are indistinguishable. In the previous point (#4), we explained how we verified that the differentiated cells expressed myogenic markers and that those do not change between the different timepoints or between stretched and non-stretched cells (Supplemental Figure 1, point #4).

These cells are plated on six-well plates with a flexible membrane to allow for cell stretching so we cannot take good quality microscopy images of these cells. However, we did check (regularly) that the morphology of the cells is similar to when we culture them on plastic and take pictures of them.

- Controls are needed to provide the stretching status of these cells by measuring the forces generated by myoblasts vs differentiated cells in response to stretch. This is very important because forces generated on myoblasts vs differentiated cells can be different even though stretching conditions are the same. The comparisons of gene expression and splicing changes among these groups in response to stretch become hazy

We agree with the reviewer that the forces generated on myoblasts would be different than the forces generated on differentiated cells because of the distinct mechanical and biological contexts of the cells. Myoblasts are actively proliferating cells that respond to a myriad of growth signals. Differentiated cells have exited the cell cycle and begun to fuse together to create long, multinucleated myotubes. Thus, we purposefully did not compare gene expression or splicing changes between myoblasts and differentiated cells. We have added some clarification about this in the Discussion section (**page 35**).

- 7. To accurately and meaningfully compare these datasets, there should be a normalization standard for differentiation and stretching “force generated” status of these cells to be able to conclude that these changes are mechanosensitive. The effectiveness of stretch in each sample within each group needs to be validated by checking markers for stretch or some other means. There is variation among samples even within same group**

The Flexcell system is unable to measure forces generated by cells. However, according to the Flexcell technical reports, 16% percent of stretch equates to 0.16 strain and -69.5 pressure (kPa) exerted on the flexible membrane that the cells are plated on. We have added this information to our manuscript (**page 5**).

As the reviewer pointed out, we did have cell stretching readouts to regularly check that the cells were effectively stretched among samples and conditions: the connective tissue growth factor (Ctgf) and the cysteine-rich angiogenic inducer 61 (Cyr61). CTGF is a matrix protein extensively involved in cell signaling. CYR61 is an early growth factor that can bind to integrins. Ctgf and Cyr61 transcripts are highly upregulated in diseases where the body fails to respond properly to mechanical stress and in response to exercise or cell stretching (Chaour and Goppelt-Struebe, **FEBS Journal**, 2006, PMID: 16856934). We have added some citations supporting the use of Ctgf and Cyr61 as our readouts for stretching (**page 7**).

Citations:

Chaour and Goppelt-Struebe. **FEBS Journal**, “Mechanical regulation of the Cyr61/CCN1 and CTGF/CCN2 proteins.” 2006

- 8. Which genes are stretch sensitive? These should be the genes that are commonly affected in all stretched cells independent of the differentiation status of the cells. I did not see this analysis in the paper. Ideally stretch sensitive genes undergo changes in response to stretch regardless of the cell differentiation status.**

To perform the analysis that the reviewer is suggesting, we overlapped the mechanosensitive genes at the transcriptional (gene expression) and splicing levels in myoblasts and differentiated cells to determine those in common between the two differentiation statuses. Overall, there was more overlap in the mechanosensitive genes for myoblasts and differentiated cells after 1 h of cell stretching than 3 h or 6 h. For splicing, there was minimal overlap between mechanosensitive splicing events between myoblasts and differentiated cells. We have incorporated this data in the supplemental figure (**Supplemental Figure 8**) and added the following paragraph in the manuscript (**pages 24-25**):

“We overlapped the mechanosensitive genes both at the transcriptional (gene expression) and splicing levels between myoblasts and differentiated cells at each timepoint after stretching. We found that after 1 h of stretching, myoblasts and differentiated cells shared numerous mechanosensitive genes (57 genes); this overlap decreased as cells were stretched longer (37 for 3 h and 43 for 6 h of stretching)

(Supplemental Figure 8a). Overall, there were few genes that were alternatively spliced that overlapped between myoblasts and differentiated cells at any timepoint (12-19 genes). Further examination of the mechanosensitive genes in both myoblasts and differentiated cells after 1 h of stretching reveals that several of them were involved in the MAPK pathway (Supplemental Figure 8b, green). Overall, there were few genes that were alternatively spliced that overlapped between myoblasts and differentiated cells at any timepoint (12-19 genes).”

In both myoblasts and differentiated cells we assayed gene expression of *Ctgf* and *Cyr61* mRNA, which were both in the overlap of gene expression (1 h and 3 h) in myoblasts and differentiated cells. In myoblasts, *Ctgf* and *Cyr61* were significantly upregulated after 1 h of stretching and returned to baseline after 3 h and 6 h (Supplementary Figure 2). In differentiated cells, *Ctgf* and *Cyr61* were significantly upregulated after 1 h and 3 h of stretching (Supplementary Figure 3). We hypothesize that we do not see changes in *Ctgf* and *Cyr61* after 6 h of stretching because the cells become accustomed to the mechanical stimulus as has been described before (Chaqour and Goppelt-Struebe, *FEBS Journal*, 2006, PMID: 16856934).

Citations:

Chaqour and Goppelt-Struebe. *FEBS Journal*, “Mechanical regulation of the Cyr61/CCN1 and CTGF/CCN2 proteins.” 2006

b

Gene symbol	Gene name
Fos	Fos proto-oncogene
Ctgf	Connective tissue growth factor
Cyr61	Cysteine-rich angiogenic inducer 1
Atf3	Activating transcription factor 3
Bmf	Bcl2 modifying factor
Dusp1	Dual specificity phosphatase 1
Egr1	Early growth response 1
Nr4a1	Nuclear receptor subfamily 4 group A member 1
Stat2	Signal transducer and activator of transcription 2
Sgk1	Serume/glucocorticoid regulated kinase 1
Mylip	Myosin regulatory light chain protein
Hoxa3	Homeobox A3

SUPPLEMENTAL FIGURE 8

Supplemental Figure 8. Overlap between mechanosensitive genes at the transcriptional (gene expression) and splicing levels between myoblasts and differentiated cells. a. The mechanosensitive genes (both at the transcriptional and splicing levels) in myoblasts and differentiated cells were overlapped. b. Examples of some of the mechanosensitive genes (1 h stretching). Genes labeled in green encode proteins that are involved in the MAPK pathway.

9. Authors mention that results from cell stretching is controversial (length and level of stretching and changes on muscle differentiation or proliferation). It is not well justified why authors chose this level and length of stretching in muscle cells. / What is the rationale of stretching 1h, 3h and 6h? Are these physiological levels? They mention something about stress being induced due to stretching. Do these conditions represent pathological conditions? Authors mention in the intro that the stretching is a part of contraction of skeletal muscle. How much force is generated after 1h, 3h, 6h of stretching on myoblasts vs differentiated myotubes?

Various groups have stretched C2C12 myoblasts and differentiated cells and focused their studies on short timepoints to study acute changes due to cell stretching (reviewed in Wang et al. *Stem Cells and Development*, 2020, PMID: 31950873). Based on the literature, we chose 1 h, 3 h, and 6 h of cell stretching to characterize an acute response (1 h) as well as a sustained response (6 h) to stretching (and an intermediate one, 3 h). We have added more information about our rationale to determine our stretching parameters (page 5).

We cannot extrapolate our *in vitro* parameters with physiological parameters. Individual cells plated in a dish are very different from a functional tissue, with multiple cell types, tridimensional structure, extracellular components, and autocrine and paracrine signals. Stretching experiments in tissue often do not parse direct, mechanosensitive effects because there can be secondary effects of stretching that impact the physiology of the mammal, whether mouse or human. Mice of different strains exhibit vastly different responses to exercise establishing a strong genetic component underlying exercise response (Avila, Kim, and Massett, *Front Physiol*, 2017, PMID: 29249981). Even individuals who undergo the same strict exercise protocol exhibit heterogenous responses and this variability has a strong genetic component (Egan and Zierath, *Cell Metabolism*, 2013, PMID: 23395166. Bouchard et al. *J Appl Physiol*, 1999, PMID: 10484570). This variability in the response in mammals establishes that it is important to examine the basic, molecular mechanisms of cell stretching in muscle cells. Further, limited studies in muscle tissue have examined the effect of cell stretching on alternative splicing. More studies need to be performed to link these two ideas – ours is one of the first. We have added some information in the Discussion expanding on the limitations of our studies and its physiological impact (page 38).

As mentioned in point #7, the Flexcell system can calculate the amount of strain on the membrane that the cells are plated on, but the system cannot calculate the force generated on or by the cells. Since myoblasts and differentiated cells exhibit different mechanical environments (proliferative vs differentiation) we do expect that there would be different responses on the cells due to the same strain. This is one reason why we do not compare myoblasts to differentiated cells in the paper. We have added some clarification in the Discussion (page 35).

Citations:

Egan and Zierath, *Cell Metabolism*, “Exercise metabolism and the molecular regulation of skeletal muscle adaptation.” 2013.

Bouchard et al. *J Appl Physiol*, “Familial aggregation of V_{O2max} response to exercise training: results from the HERITAGE Family Study.” 1999.

Wang et al. *Stem Cells and Development*, “Multiple effects of mechanical stretch on myogenic progenitor cells.” 2020.

10. Focusing on a specific target gene relevant to muscle disease or muscle function is necessary to put a perspective on the significance of these findings and to the statement that these genes are mechanosensitive. The conclusions of this paper are based on GO analysis of gene expression and splicing changes. GO analysis is a good way to find out overall processed affected but there could be other important changes in essential genes (only a few) that do not necessarily form a group in GO analysis. It is important to assess the consequences of these changes on muscle cell physiology.

We agree with the reviewer on this point and added some more information about two genes (*Fos* and *Ctgf*) that undergo strong gene expression changes after 1 h and 3 h of stretching in differentiated cells and are both involved in the MAPK pathway. We discuss subsequent functional implications of these genes in the Discussion (**page 36**):

“Further, some of the mechanosensitive genes after 1 h of stretching (myoblasts and differentiated cells) are activated by the MAPK pathway. *Fos* is an early response gene that is highly upregulated in response to exercise and involved in a wide variety of cellular processes including transcription⁵⁰. *Ctgf* is another exercise early response gene and CTGF is recruited to muscle fibers likely to promote remodeling of matrix proteins²⁷. The strong activation of these two genes after 1 h of stretching in both myoblasts and differentiated cells suggests a role of the MAPK pathway in regulating the stretch response through transcriptional programs.”

11. Myoblasts represent early developmental stages. But same stretching conditions were used for both myoblasts and differentiated cells that represent adult stages. The stretches applied to these cells might not be the same as the muscle tissue at embryo vs adult stages. This should be addressed.

We agree that stretching myoblasts may have a different effect than stretching differentiated cells. It is known that myoblasts and differentiated cells have distinct mechanical environments. Our goal was not to compare myoblasts and differentiated cells. Thus, we opted to apply standard stretching conditions to both myoblasts and differentiated cells.

12. It is important that authors overexpress WT or non-phosphorylatable SRSF4 in myoblasts or differentiated cells to examine how global splicing is affected and to test if these cells respond to stretch similarly

We thank the reviewer for this interesting suggestion. C2C12 cells are notoriously difficult to transfect with plasmids for overexpression experiments. Further, SRSF4 has the largest RS domain out of all the SR proteins indicating that it has many potential phosphorylation sites. Thus, it would not be feasible to create a non-phosphorylatable SRSF4 because it would likely affect the ability of the protein to function

and fold correctly. Even though we appreciate this very interesting suggestion, we respectfully think that it is outside the scope of the present manuscript.

13. To determine direct targets of SRSF4 affected in stretched muscle cells via splicing, it would be better to perform SRSF4-CLIP sequencing analysis or use available SRSF4 CLIP-seq datasets from ENCODE project.

We found this comment to be very exciting. We utilized an available iCLIP dataset (there were no CLIP datasets; iCLIP identifies cross-link sites by providing individual nucleotide resolution) that examined SRSF4 binding sites in mouse embryonic carcinoma cells (P19) (Änkö et al *Genome Biol*, 2012, PMID: 22436691). We found that there were few mechanosensitive genes (both at the transcriptional and/or splicing levels) that overlapped with the genes bound by SRSF4 (those containing iCLIP tags) (**Image 3 for response to reviewers**). It is expected that the mouse embryonic carcinoma cells used in that iCLIP experiment have a very different mechanical context than mouse skeletal muscle cells. That is likely one reason why we do not observe many overlapping genes. It would be interesting in a future paper to perform SRSF4-CLIP sequencing analysis on skeletal muscle cells but that is out of scope for this current paper.

Citations:

Änkö et al. *Genome Biol*, “The RNA-binding landscapes of two SR proteins reveal unique functions and binding to diverse RNA classes.” 2012.

Image 3 for response to reviewers. SRSF4 iCLIP targets overlapped with mechanosensitive genes and splicing events. SRSF4 iCLIP protein-coding targets were overlapped with the mechanosensitive genes and splicing events from myoblasts and differentiated cells.

14. How is SRSF4 regulated in response to stretch in muscle cells? This is not thoroughly addressed. SRSF4 levels are increased according to the RNA-seq data. What’s the reason for increased SRSF4 mRNA levels?

According to the heatmap in **Figure 6a**, SRSF4 mRNA levels are not upregulated in response to stretch in muscle cells. We also performed a validation by quantitative real time PCR (qPCR) of SRSF4 mRNA expression which confirmed that SRSF4 mRNA expression does not respond to cell stretching (**Image 4 for response to reviewers**). In **Figure 7a**, it is clear that SRSF4 protein is also not changing due to cell stretching (western blotting).

Image 4 for response to reviewers. SRSF4 mRNA expression after stretching differentiated cells. SRSF4 mRNA expression was evaluated by qPCR. The stretched samples were normalized to the non-stretched samples from the same time point. Results are shown as mean \pm SEM, $N=4$.

15. SRSF4 protein phosphorylation is increased. Is phosphorylation of SRSF4 MAPK/ERK dependent?

Figure 7a shows that after 1 h of stretching SRSF4 protein phosphorylation is decreased (not increased) even though the total SRSF4 protein does not change. To evaluate if this response is ERK dependent, we utilized the ERK phosphorylation inhibitor, U0126, and stretched cells for 1 h. We observed that the ERK inhibitor did not affect SRSF4 phosphorylation in basal conditions (compare lane 1 and 3 in **Image 5 for response to reviewers**) but resulted in a very small trend for less SRSF4 phosphorylation in response to stretching (**Image 5 for response to reviewers**). This indicates that phosphorylation of SRSF4 is not downstream of ERK phosphorylation in basal conditions. Since the reduction of SRSF4 phosphorylation in response to stretching slightly responded to ERK phosphorylation inhibition this suggests that phosphorylated ERK is a negative inhibitor of SRSF4 phosphorylation. Typically, phosphorylated ERK promotes phosphorylation of other proteins so this result was surprising and indicates that there may be other proteins in the signaling pathway between ERK and SRSF4.

Image 5 for response to reviewers. Effect of ERK inhibition on the phosphorylation of SRSF4. Differentiated cells were treated with the ERK phosphorylation inhibitor U0126 (30 min) and then stretched for 1 h. Western blot assays were performed and quantified by densitometry to examine phosphorylated levels of SRSF4 (p-SRSF4). Results are shown as mean \pm SEM, $N=3$. * $p<0.05$, Welch's t-test.

16. Figure 5: There is unusually high number of out of frame splicing changes. This should be discussed why? Are there any examples of this in muscle diseases or in other conditions?

We recognized that there was an error in our method of calculating the length of the alternative exon. After correcting for this error, 48%, 61%, and 50% of cassette exons (1 h, 3 h, 6 h, respectively) were in frame transitions for myoblasts and 48%, 45%, and 43% of cassette exons (1 h, 3 h, 6 h, respectively) were in frame transitions for differentiated cells. This suggests that there are about equal numbers of in frame and out of frame events for both myoblasts and differentiated cells. This was corrected in **Figure 5**.

In Duchenne's muscular dystrophy (DMD), there are numerous instances of genomic deletions that alter the reading frame leading to the production of truncated proteins (Singh and Cooper, *Trends Mol Med* 2012, PMID: 22819011. Douglas and Wood, *Mol Cell Neurosci* 2013, PMID: 23631896). Several of the current molecular therapies for DMD involve the use of antisense oligonucleotides to help restore the correct reading frame.

Citations:

Douglas and Wood. *Mol Cell Neurosci*, "Splicing therapy for neuromuscular disease." 2013.

Singh and Cooper. *Trends Mol Med*, "Pre mRNA splicing in disease and therapeutics." 2012.

17. Figure 6A: Authors state that DNA damage is affected based on GO analysis. If DNA damage is present in response to stretch, this should be validated in these cells upon stretch at different time points. GO analysis is not adequate for such a conclusion. This is important because DNA damage can cause apoptosis.

We thank the reviewer for this suggestion and comment of how to frame some of our statements. We agree that GO analysis is not at all enough for this type of conclusions. We think that assaying DNA damage is important. The GO analysis indicated that genes encoding proteins involved in DNA damage response were significantly alternatively spliced after stretching. We stretched differentiated cells for 6 h to examine if alterations to the DNA damage response were occurring. We utilized H2AX phosphorylation compared to total H2AX as the readout. Phosphorylation of H2AX is an early response to DNA double strand breaks so this method is considered a robust marker to determine DNA damage (Mah et al. *Nature*, 2010 PMID: 20130602).

We observed that the stretched cells exhibited less phosphorylated H2AX indicating that there was less DNA damage than basal levels (**Image 6 for response to reviewers**). A recent paper demonstrated that NIH3T3 fibroblasts exposed to UV radiation and stretched for 12 h exhibited less DNA damage than fibroblasts just exposed to UV radiation (Nagayama et al. *Biomechanics and Modeling in Mechanobiology*, 2020, PMID: 31506862). Further, that work showed that NIH3T3 fibroblasts stretched for 12 h exhibited less DNA damage than non-stretched cells. The authors hypothesized that this is due to chromatin condensation that happens after cells are exposed to stretching which has also been described by other groups (Irianto et al. *Biophysical Journal*, 2017, PMID:28341535).

Citations:

Mah et al. *Nature*, "gammaH2AX: a sensitive molecular marker of DNA damage and repair." 2010.

Nagayama et al. *Biomechanics and Modeling in Mechanobiology*, "Cyclic stretch-induced mechanical stress to the cell nucleus inhibits ultraviolet radiation-induced DNA damage." 2020.

Irianto et al. *Biophysical Journal*, “As a nucleus enters a small pore, chromatin stretches and maintains integrity, even with DNA breaks.” 2017.

Image 6 for response to reviewers. Stretching cells for 6 h results in less H2AX phosphorylation. Differentiated cells were stretched for 6 h and protein was immediately harvested. Western blot assays were performed and quantified by densitometry to examine total and phosphorylated levels of H2AX (p-H2AFX). Results are shown as mean \pm SEM, $N=3$. * $p<0.05$, Welch’s t-test.

18. Figure 1B: There is no obvious clustering based on time point or stretch vs non-stretched cells

We thank the reviewer for their comment and realize that ‘clustering’ may be a misleading term for PCA analysis. Thus, we have adjusted the text in the manuscript on **pages 5-6** to indicate that we observe segregation of the samples since PCA analysis allows us to test for segregation along each PC.

PCA was used to decompose the myoblast and differentiated cells gene expression matrix. The sample loadings derived for PC1 and PC2 were tested for association with stretch with a t-test or treatment conditions with an ANOVA. There was significant segregation of the myoblast samples by treatment (stretch or no stretch) in PC2 ($p<0.05$) and by time (1 h, 3 h, or 6 h) in PC1 and PC2 ($p<0.05$). We also performed segregation analysis of the genes in differentiated cells and observed significant segregation by treatment (stretch or no stretch) in PC2 ($p<0.05$) and by time (1 h, 3 h or 6 h) in PC1 ($p<0.05$).

Overall, this indicates that the individual samples exhibit different gene expression changes that are captured by the experimental conditions of stretch/no stretch and time of stretching (1 h, 3 h, or 6 h) for both myoblasts and differentiated cells.

19. Fig 1E & F: Criteria for selecting these genes to show correlation is not explained

We selected genes for validation with a fold change > 1.5 for upregulated genes and a fold change < -1.5 for downregulated genes and an adjusted p value < 0.05 . We added these criteria to **pages 6-7** The validated genes (which are shown in **Supplementary Figure 2 and Supplementary Figure 3**) were then utilized to create the correlations.

20. Fig 2: Based on the log10 value only GO term correlated to data seems to be reg. DNA-templated transcription. The other GO p-values are high to be significant.

For this analysis, we opted to use the adjusted *p*-value which is considered stricter. The value we chose is also considered appropriate for hypothesis generation for sequencing experiments (Subramanian et al. *Proc Natl Acad Sci USA*, 2005, PMID: 16199517).

Citations:

Subramanian et al. *Proc Natl Acad Sci USA*, “Gene set enrichment analysis: a knowledge-based approach for interpreting genome-wide expression profiles.” 2005.

21. Fig 4B: PC analysis does not show clustering for splicing datasets

We adjusted the text in the manuscript on **page 16** to indicate that we observe segregation of the samples since PCA analysis allows us to test for segregation along each PC.

PCA was used to decompose the myoblast and differentiated cells alternative splicing events. The sample loadings derived for PC1 and PC2 were tested for association with stretch with a t-test or treatment conditions with an ANOVA. There was significant segregation of both the myoblast and differentiated cells samples by treatment (stretch or no stretch) in PC2 ($p < 0.05$) and by time (1 h, 3 h or 6 h) in PC1 ($p < 0.05$).

Overall, this indicates that the individual samples exhibit different alterations in alternative splicing that are captured by the experimental conditions of stretch/no stretch and time of stretching (1 h, 3 h, or 6 h) for both myoblasts and differentiated cells.

22. Fig 5D- if 60-80 % of CE events are out of frame transition, does this correlate with truncated protein: or is there any evidence of mRNA degradation?

To analyze out of frame transitions we took the events with an out of frame transition (not divisible by three) and analyzed their mRNA expression levels before and after stretching to determine if there were significant changes (**Table 1 for response to reviewers**) that could be potentially due to mRNA degradation or non-sense mediated decay. There was only one case at 1 h in differentiated cells, another one at 3 h in differentiated cells, and another one at 6 h in myoblasts that were significantly changing their mRNA levels in response to stretching (in addition to changing the splicing patterns) (**Table 1 for response to reviewers**). There were no splicing events with concomitant differences in mRNA expression levels for 6 h in differentiated cells, and 1 h or 3 h in myoblasts (**Table 1 for response to reviewers**). This analysis led us to conclude that the out of frame transitions are not correlated with resulting mRNA expression changes that could have been associated with non-sense mediated decay or degradation.

	1 h stretching Myoblasts	3 h stretching Myoblasts	6 h stretching Myoblasts	1 h stretching Differentiated cells	3 h stretching Differentiated cells	6 h stretching Differentiated cells
Genes with alternative exons not divisible by 3	0	0	1	1	1	0

and fold change > 1.5						
Genes with alternative exons not divisible by 3	21	30	53	27	51	43

Table 1 for response to reviewers. Genes with exons not divisible by three do not exhibit strong gene expression changes. The genes that contained exons not divisible by three were analyzed to determine if strong gene expression changes occurred.

23. Fig6A: Log10 pvalue are too low for any conclusive GO terms

There were few significant splicing changes therefore we opted to use a less stringent (but still acceptable) log₁₀ pvalue to help generate hypotheses of what biological processes the mechanosensitive splicing events were involved in.

24. Fig8 – data indicate a correlation between the gene expression and splicing but these is not enough data to draw a causative conclusion.

We appreciate the reviewer’s comment and have clarified the text to say that we are not drawing a causative correlation but are indicating potential connections that could be further explored (**page 31**).

Minor issues:

1. The transition from Figure 1 to Figure 2 is not very smooth. The rationale for analyzing splicing is not well justified in the results section

We took the reviewer’s advice and combined Figure 1 and 2 into one figure (current **Figure 1**) which allowed for a better transition between some of the data. At the beginning of the splicing results section on **page 16**, we have a paragraph detailing how splicing can be affected by external factors and we hypothesized that mechanical stretching could induce changes in splicing similar to other external factors:

“RNA processing can be regulated by multiple factors including transcriptional speed, temperature, and cellular stress^{28–30}. Because environmental factors can alter RNA processing mechanisms, we hypothesized that mechanical stretching induces alternative splicing changes in skeletal muscle cells.”

2. Figure 5 and 6 can be combined into one figure. The rest of details can be added as a supplemental figure.

We appreciate the reviewer’s suggestion and combined Figure 5 and 6 into one figure (current **Figure 5**) and moved the splicing GO analysis and the inclusion/exclusion pie charts for all the events as a supplemental figure (**Supplemental Figure 7**).

3. Figure 1 and 2 can be combined into one figure and the rest of details can be added as a supplemental figure.

We appreciate the reviewer's suggestion and combined Figure 1 and 2 into one figure (current **Figure 1**) and moved the gene expression GO analysis as a supplemental figure (**Supplemental Figure 4**).

4. Comments on splicing changes affecting proteome without proteomic analysis should be toned down.

We thank the reviewer for this recommendation. We have toned down the comments regarding the proteome since it is not within the scope of our paper to make conclusions about proteomics.

5. Splicing events are shown as % in pie charts. It is hard to see the exact number of splicing events and mRNA level changes from the graph plots. It is important to add the total number of splicing events and how many total genes undergo splicing changes and mRNA level changes in figure legends.

We have added the exact number of splicing events to the pie charts in all the main and supplemental figures.

REVIEWER 2

1. I believe the manuscript would benefit from more context describing the mechanical loading parameters and the motivation for choosing these parameters. The authors state that 16% biaxial stretch for 1 h, 3 h and 6 h was used. Would a longer time for stretching cycle change the results in any way? Similarly, biaxial stretch does not seem to be the correct choice for muscle, as muscle fibers are loaded primarily along one axis. This context of the mechanical loading might in-fact have an important impact on the results, given that it is known that cytoskeletal reorganization will respond to the direction of mechanical loading on flexible substrates.

Our motivation for choosing the stretching times of 1 h, 3 h, and 6 h was based on both the literature and practical experience with the cells. Various groups have stretched C2C12 myoblasts and differentiated cells and focused their studies on short timepoints to study acute changes due to cell stretching (reviewed by Wang et al. *Stem Cells and Development* 2020. PMID: 31950873). Over time, cells become accustomed to the stimulus of stretching. Indeed, we have performed experiments stretching cells for 18 h (not presented in the paper). Cells were negatively affected and died. We have added more information about our determination of stretching parameters on **page 5**.

Thank you for helping us clarify our stretching axis. The Flexcell system that we used employs a radial/equibiaxial stretch which applies an equal amount of tension on the cells. We have updated the manuscript to indicate that we are applying an equibiaxial stretch and not a biaxial stretch. The myoblasts and differentiated cells do not align along one axis like mature muscle fibers do – they are more disorganized and exhibit different orientations (**Image 7a for response to reviewers**). Because of this an equibiaxial stretch is more appropriate for our cells compared to a uniaxial stretch especially given the point that cytoskeletal reorganization would only affect one side of a muscle cell in a uniaxial stretching system (**Image 7b for response to reviewers**).

The Flexcell system can employ a maximum stretch of 21.8% according to the Flexcell catalogs, however, in experimental settings we were only able to stretch the cells at a maximum of 16%. We chose to use 16%, because we wanted to stretch cells enough to induce mechanosensitive gene

expression and alternative splicing changes and because several other groups have stretched cells at about this level. A 16% stretch corresponds to a pressure of -69.5 (kPa), and 0.16 strain (**Image 7b for response to reviewers**).

Image 7 for response to reviewers. C2C12 cell organization and stretching. a. Immunofluorescence example of C2C12 differentiated cells b. Flexcell stretching parameters.

Citations:

Wang et al. *Stem Cells and Development*, "Multiple effects of mechanical stretch on myogenic progenitor cells." 2020.

2. I found it difficult to place these findings in a specific physiological context. The authors describe a broad range of possible implications for the results, including development and muscular diseases, which is very interesting, but which of these is best described by the specific mechanical loading conditions that are used in this study? For example, the authors compare no stretch with 1 h, 3 h and 6 h of stretching, but adult muscle presumably is subject to constant cyclic strain (for example in the heart). This again implies that an even longer time for the loading protocol might be more appropriate, and a higher strain magnitude.

While heart and skeletal muscle are both striated muscles, the heart does undergo unique cyclic stretching every time it beats. Adult skeletal muscle is subject to repeated stretching in workout settings but otherwise undergoes intermittent stretching. It is difficult to extrapolate *in vitro* parameters with physiological parameters because muscle cells grown in a dish have a different mechanical context than muscle tissue. Further, physiological studies have limitations because stretching a muscle tissue may result in secondary effects based on the physiology of an animal.

The goal of our study was to molecularly define direct gene expression and splicing targets of stretching without potential confounding secondary effects. The mechanosensitive genes and splicing events that we identified would be great candidates to follow up on in muscle tissue or diseased muscle. Overall, more molecular studies need to be performed to help inform pathological research in development and muscular diseases.

We have added some limitations of our study to the Discussion to help situate our results within the field (**page 38**). We think it is important to share potential implications of our results, but we recognize that we must not overstate our claims. Thank you for helping us improve our work.

REVIEWER 3

1. In Fig.2, it would be interesting to add a table or Venn diagram showing what exactly are the genes that overlap between the myoblasts and the differentiated cells.

To perform the analysis that the reviewer is suggesting, we overlapped the mechanosensitive genes at the transcriptional (gene expression) and splicing levels in myoblasts and differentiated cells to determine those in common between the two differentiation statuses. Overall, there was more overlap in the mechanosensitive genes from myoblasts and differentiated cells after 1 h of cell stretching than 3 h or 6 h. For splicing, there was minimal overlap between mechanosensitive splicing events between myoblasts and differentiated cells. We have incorporated this data in the **Supplemental Figure 8** and added the following paragraph in the manuscript (**pages 24-25**):

“We overlapped the mechanosensitive genes both at the transcriptional (gene expression) and splicing levels between myoblasts and differentiated cells at each timepoint after stretching. We found that after 1 h of stretching, myoblasts and differentiated cells shared numerous mechanosensitive genes (57 genes); this overlap decreased as cells were stretched longer (37 for 3 h and 43 for 6 h of stretching) (**Supplemental Figure 8a**). Overall, there were few genes that were alternatively spliced that overlapped between myoblasts and differentiated cells at any timepoint (12-19 genes). Further examination of the mechanosensitive genes in both myoblasts and differentiated cells after 1 h of stretching reveals that several of them were involved in the MAPK pathway (**Supplemental Figure 8b, green**). Overall, there were few genes that were alternatively spliced that overlapped between myoblasts and differentiated cells at any timepoint (12-19 genes).”

b

Gene symbol	Gene name
Fos	Fos proto-oncogene
Ctgf	Connective tissue growth factor
Cyr61	Cysteine-rich angiogenic inducer 1
Atf3	Activating transcription factor 3
Bmf	Bcl2 modifying factor
Dusp1	Dual specificity phosphatase 1
Egr1	Early growth response 1
Nr4a1	Nuclear receptor subfamily 4 group A member 1
Stat2	Signal transducer and activator of transcription 2
Sgk1	Serume/glucocorticoid regulated kinase 1
Myh9	Myosin regulatory light chain protein
Hoxa3	Homeobox A3

SUPPLEMENTAL FIGURE 8

Supplemental Figure 8. Overlap between mechanosensitive genes at the transcriptional (gene expression) and splicing levels between myoblasts and differentiated cells. **a.** The mechanosensitive genes (both at the transcriptional and splicing levels) in myoblasts and differentiated cells were overlapped. **b.** Examples of some of the mechanosensitive genes (1 h stretching). Genes labeled in green encode proteins that are involved in the MAPK pathway.

- In Fig.5, the authors show that cassette exons that are divisible by three are more common in differentiated cells over time and suggest that as differentiated cells are stretched longer, there will be more instances of proteins having a defined, internal sequence added to them. However, this does not consider that exons divisible by three may also carry translational stop codons. Can this be analyzed in any way?**

We recognized that there was an error in our method of calculating the length of the alternative exon. After correcting for this error, 48%, 61%, and 50% of cassette exons (1 h, 3 h, 6 h, respectively) were in frame transitions for myoblasts and 48%, 45%, and 43% of cassette exons (1 h, 3 h, 6 h, respectively) were in frame transitions for differentiated cells. This suggests that there are about equal numbers of in frame and out of frame events for both myoblasts and differentiated cells. This was corrected in **Figure 5**.

To know exactly if in each of the proteins there is an insertion of a stop codon, we would need to go one by one and analyze the potential transcripts based on the information available (annotated transcripts in different databases). This is not feasible to do. However, we performed analysis to determine if in the alternative exons that were divisible by three there were stop codons present (we had to evaluate each of the three potential reading frames). This is what we found in differentiated cells:

- After 1 h of stretching: 50% of the exons in reading frame #1 and #3 had a stop codon present and 42% of the exons in reading frame #2 had a stop codon present.

- After 3 h of stretching: similar numbers were observed

- After 6 h of stretching: reading frame #2 and #3 exhibited very different numbers of exons with stop codons present in them.

Therefore, as the reviewer suggested it is possible that the alternative exons that are divisible by three could be adding a stop codon thus causing protein truncation. In those cases, functional consequences are highly probable.

Image 8 for response to reviewers. Mechanosensitive exons divisible by three can produce truncated proteins due to stretching. The mechanosensitive exons divisible by three were analyzed in the three potential reading frames to determine if stop codons were present in the exons. The numbers between parentheses indicate the number of cassette exons.

3. In Figure 6, what is the correlation (if any) between the genes that are alternatively spliced and those that were transcriptionally regulated? Has this been looked at?

Overall, there is very little overlap between the genes that are alternatively spliced and the genes that are differentially expressed. This suggests that the splicing changes occurring due to stretching result in altered protein isoforms as opposed to variation in gene expression. This phenomenon has previously been characterized in numerous different contexts (Giudice et al. *Nature Communications*, 2014, PMID: 24752171; Parikshak et al. *Nature*, 2016, PMID: 29995847; Li et al. *Science*, 2016, PMID: 27126046; Jacobs et al. *Molecular Ecology*, 2021, PMID: 33502030). We have added this data as a supplemental figure (**Supplemental Figure 9**) and added the following paragraph about the data on **page 25**:

“We overlapped the mechanosensitive genes at the transcriptional (gene expression) level with genes changing their alternative splicing patterns in response to stretching at each timepoint for myoblasts and differentiated cells (**Supplemental Fig. 9**). We found that there was a minimal overlap between the mechanosensitive genes with altered transcription and those changing their splicing patterns (**Supplemental Fig. 9**). This indicates that transcriptional changes and splicing variation due to cell stretching are involved in distinct cellular processes and are independent of each other.”

SUPPLEMENTAL FIGURE 9

Supplemental Fig 9. Genes that are mechanosensitive at the transcriptional level are distinct from those changing their splicing patterns. The mechanosensitive genes at the transcriptional level were overlapped with those changing their alternative splicing patterns in response to stretching for each time point (1 h, 3 h, 6 h) and cellular condition (myoblasts, differentiated cells).

Citations:

Giudice et al. *Nature Communications*, “Alternative splicing regulates vesicular trafficking genes in cardiomyocytes during postnatal heart development.” 2014.

Parikshak et al. *Nature*, “Genome-wide changes in lncRNA, splicing, and regional gene expression patterns in autism.” 2016.

Li et al. *Science*, “RNA splicing is a primary link between genetic variation and disease.” 2016.

Jacobs et al. *Molecular Ecology*, “Alternative splicing and gene expression play contrasting roles in the parallel phenotypic evolution of a salmonid fish.” 2021

4. In Figure 7, what is the overlap of the results of SRSF2, SRSF5, and SRSF6 are pooled together and compared to the stretched cells? Have the authors tried different combinations to see which pooled results cover the highest proportion of observed changes?

Based on the reviewer’s advice we performed an analysis where we pooled the genes that responded to SR protein overexpression (at the transcriptional and splicing level) with those that we found to be mechanosensitive. Approximately 20% of the transcriptional targets of SRSF2, SRSF4, or SRSF6 overlapped with the mechanosensitive genes (**Image 9 for response to reviewers, left**). 34% of the cassette exons that respond to overexpression of SRSF2, SRSF4, SRSF6, or SRSF9 overlapped with cassette exons that respond to cell stretching (**Image 9 for response to reviewers, right**). These percentages are higher than those of individual SR protein pie charts in **Fig. 6b-c**, but comparing the transcriptional and splicing targets of the individual SR proteins to stretch-responsive genes is more informative. That allows us to determine that SRSF4 and SRSF6 targets overlapped with mechanosensitive genes and splicing events more so than the other SR proteins (**Fig. 6b-c**).

Image 9 for response to reviewers. SR protein targets overlapped between mechanosensitive genes at the transcriptional and splicing levels. The SR protein (SRSF2, SRSF4, and SRSF6) transcriptional targets were overlapped with mechanosensitive genes (transcriptional level), *left*. The cassette exons that respond to SR protein overexpression (SRSF2, SRSF4, SRSF6, and SRSF9) were overlapped with the alternative cassette exons that respond to cell stretching (splicing level), *right*. The numbers between parentheses indicate the number of genes or splicing events.

Reviewer #1 (Remarks to the Author):

Thanks to the authors addressing my concerns. I recommend that authors include Image 2 in rebuttal letter as a supplemental figure in the paper (WB of ERK phosphorylation during C2C12 differentiation). The change is so dramatic that should be communicated with the readers. I have no other concerns.

Reviewer #2 (Remarks to the Author):

I appreciate the authors responses to my questions and their clearly presented rebuttal letter. I believe this manuscript is now suitable for publication.

Reviewer #3 (Remarks to the Author):

The author has answered well all the queries from this reviewer

REVIEWER 1

Major Comments:

1. Thanks to the authors addressing my concerns. I recommend that authors include Image 2 in rebuttal letter as a supplemental figure in the paper (WB of ERK phosphorylation during C2C12 differentiation). The change is so dramatic that should be communicated with the readers. I have no other concerns.

We thank the reviewer for their suggestion and we have included this image as **Supplemental Figure 5** and added the following sentence about the figure in the main manuscript on **pages 8-9**. We opted to show only one replicate of the blot to match the other figures in the paper.

“ERK1/2 phosphorylation is drastically decreased during C2C12 cell differentiation while total ERK1/2 protein levels did not change (**Supplemental Fig. 5**).”